# WORLDPM: UNDERSTANDING SCALING PATTERNS IN HUMAN PREFERENCE MODELING

## ABSTRACT

Motivated by scaling laws in language modeling that demonstrate how test loss scales as a power law with model and dataset sizes, we find that similar laws exist in preference modeling. We propose **World Preference Modeling** (WorldPM) to emphasize this scaling potential, where World Preference embodies a unified representation of human preferences. In this paper, we collect preference data from public forums covering diverse user communities, and conduct extensive training using 15M-scale data across models ranging from 1.5B to 72B parameters. We observe distinct patterns across different evaluation metrics: (1) Adversarial metrics (ability to identify deceptive features) consistently scale up with increased training data and base model size; (2) Objective metrics (objective knowledge with well-defined answers) show emergent behavior in larger language models, highlighting WorldPM's scalability potential; (3) Subjective metrics (subjective preferences from a limited number of humans or AI) do not demonstrate scaling trends. Further experiments validate the effectiveness of WorldPM as a foundation for preference fine-tuning. Through evaluations on 7 benchmarks with 16 subtasks, we find that WorldPM broadly improves the generalization performance across human preference datasets of varying sizes (7K, 100K and 800K samples), with performance gains exceeding 5% on many key subtasks.

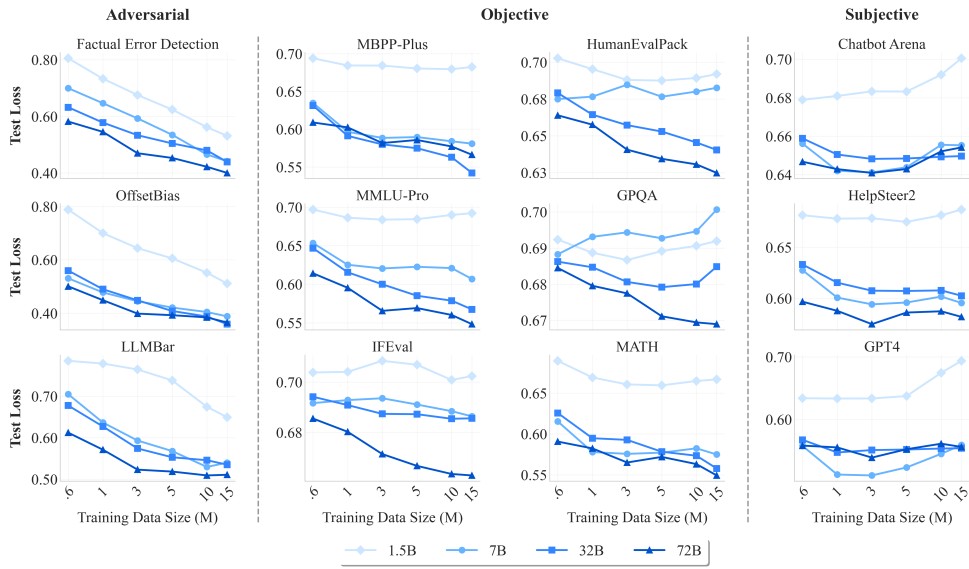

Figure 1: Test loss analysis across domains for 1.5B-72B models with increasing training data. The x-axis is scaled logarithmically (base 2).

# 1 INTRODUCTION

Kaplan et al. (2020) demonstrates that language modeling performance exhibits scaling laws, where cross-entropy loss scales as a power law with training scale. These laws have achieved remarkable success in next-token prediction tasks, continuously advancing the capabilities of language models (Touvron et al., 2023; Yang et al., 2024a;b). We argue that such scaling laws can be extended to preference modeling and thus propose World Preference Modeling (WorldPM), aiming to achieve unified preference representations through large-scale preference modeling, similar to unified representations in language modeling. A key challenge to this investigation lies in the scalability constraints of manually annotated preference data (Wang et al., 2024c; Bai et al., 2022b).

As platforms that facilitate large-scale interaction and opinion sharing, online forums aggregate vast amounts of user preference data through voting mechanisms, substantially advancing research on collective decision making in social sciences and psychology (Easley et al., 2010; Pal et al., 2012; Hu et al., 2008). Despite the immense value of these preference signals, they have rarely been effectively leveraged at scale by artificial intelligence. In one of the few efforts, Askell et al. (2021) utilized forum data for Preference Model Pre-training (PMP). However, they mainly focus on how PMP improves sample efficiency during the preference fine-tuning process, leaving the scaling properties of preference pre-training itself unexplored.

We first collect preference data from public forums, including StackExchange, Reddit, and Quora. After assessing the data quality, we conduct large-scale preference training with 15 million training samples from StackExchange on language models with parameters ranging from 1.5B to 72B. Subsequently, we evaluate the test loss across various model sizes and training scales using 12 test sets, broadly classified into adversarial, objective, and subjective categories based on the capabilities they assess. The main results in Figure 1 prove that WorldPM displays distinct scaling patterns across different domains. Specifically:

(1) **In adversarial evaluation, test losses demonstrate a power law decrease**, underscoring the model's enhanced ability to detect responses containing intentional errors, as well as those that are well-written but irrelevant or incomplete. This finding suggests that subtle deficiencies commonly plaguing discriminative models (Park et al., 2024; Zeng et al., 2024) can be effectively mitigated through large-scale training.

(2) **The objective evaluation reveal an emergent scaling phenomenon**, where larger models demonstrate a power law reduction in test losses across more benchmarks, whereas smaller models show limited or no such improvements. These results suggest that WorldPM represents a challenging task that requires larger models to elicit preferences for objective knowledge, pointing to its substantial potential for further advancement.

(3) **Subjective evaluations show no scaling trends**. We analyze potential reasons from the perspective of style preferences (e.g., preference for longer or richer responses). While WorldPM becomes more style-neutral as it scales up, some subjective evaluations exhibit style preferences, resulting in lower evaluation performance. Similar phenomena have been observed in many other studies (Dubois et al., 2024; Chiang et al., 2024).

We explore WorldPM's potential as a foundation model for preference fine-tuning. Unlike Askell et al. (2021)'s initial work that was limited to small datasets and in-distribution testing, we conduct comprehensive experiments using three human preference datasets (7K, 100K and 800K samples) and evaluate the model on multiple benchmarks. Results show that WorldPM significantly improves preference fine-tuning generalization, with performance gains over 5% on many subtasks. Finally, we reveal a positive correlation between the scale of WorldPM and the benefits obtained from fine-tuning, indicating scalable advantages as training scale increases.

# 2 BACKGROUND & RELATED WORK

## 2.1 REINFORCEMENT LEARNING FROM HUMAN FEEDBACK

Reinforcement Learning from Human Feedback (RLHF) has emerged as a critical technique for aligning large language models (LLMs) with human preferences (Christiano et al., 2017; Stiennon et al., 2020; Ouyang et al., 2022; Dubey et al., 2024; Yang et al., 2024a;b). The framework typically

involves two key stages: *reward modeling* and *reinforcement learning optimization* (Bai et al., 2022a; Zheng et al., 2023b). As it focuses on human preferences, reward modeling is also known as preference modeling (PM) (Ouyang et al., 2022). We use both terms interchangeably in this work.

A preference model (PM) learns from pairwise comparison data $(x, y_0, y_1)$, where label $Y \in \{0, 1\}$ indicates which response is preferred. The model computes scores $r_\theta(x, y_0)$ and $r_\theta(x, y_1)$ for each response using reward function $r_\theta$. Following the Bradley-Terry model (Bradley and Terry, 1952), the preference probability is:

$$P(Y = 0|x, y_0, y_1) = \text{sigmoid}(r_\theta(x, y_0) - r_\theta(x, y_1)), \tag{1}$$

The PM replaces the LLM's last decoding layer with a linear layer to output a scalar score and is trained on preference dataset $\mathcal{D}$ to minimize the Bradley-Terry (BT) loss (Sun et al., 2024):

$$\mathcal{L}_{BT} = -\mathbb{E}_{(x,y_0,y_1,Y)\sim\mathcal{D}}[\log P(Y|x, y_0, y_1)], \tag{2}$$

The trained PM guides policy optimization through RL algorithms like PPO (Ouyang et al., 2022) to maximize scores. However, RL-based alignment faces challenges in comparing different PMs due to its complex fine-tuning process, leading to stability and reproducibility issues (Sun et al., 2024). Best-of-N (BoN) sampling provides a simpler, fine-tuning-free alternative (Xu et al., 2024; Gao et al., 2023). BoN works by first sampling N responses from a model for a given prompt, then using a PM to rank these responses, and finally selecting the highest-scoring one. By avoiding complex RL optimization, BoN achieves better performance while maintaining stability and reproducibility through direct ranking (Gao et al., 2023; Coste et al., 2023; Ivison et al., 2024; Li et al., 2023).

## 2.2 Preference Modeling

Existing preference modeling and alignment methods can be primarily divided into two categories: The first category is scalar reward modeling based on the Bradley-Terry (BT) model. The second category involves text-based generative reward modeling methods like GenRM and LLM-as-judge (Findeis et al., 2024; Liu et al., 2025; Chen et al., 2025; Gu et al., 2025). WorldPM focuses on the BT scalar model and investigates its scalability. Previous research often utilized small-scale, high-quality human-annotated datasets (typically containing $10^3$–$10^5$ examples) (Wang et al., 2025; Cui et al., 2023) or larger AI-annotated datasets that may exhibit systematic biases (Chiang and yi Lee, 2023; Fu and Liu, 2025). In contrast, WorldPM leverages large-scale forum data, which inherently contains human preference signals to empirically study scaling laws in reward modeling.

Another line of research examines the vulnerabilities of reward models during policy optimization and evaluation. With a fixed reward model, enhanced optimization often leads to overoptimization and Goodhart-like effects (Gao et al., 2023). Additionally, reward models and LLM-as-judge systems exhibit biases in verbosity, self-preference, and adversarial settings (Spiliopoulou et al., 2025; Chen et al., 2024), prompting research into debiasing and calibration techniques (Choi et al., 2025; Satterfield et al., 2025; Dai et al., 2025; Wolf et al., 2025). These studies typically fix the reward model or evaluator and adjust only optimization procedures or inputs. In contrast, WorldPM demonstrates that expanding the training scale can enhance objective discrimination and reduce biases.

## 2.3 Preference Model Pre-training

Due to the high cost of preference annotation, Askell et al. (2021) propose using public forum data for Preference Model Pre-training (PMP). Our work extends theirs in two key aspects: (1) we analyze the scaling properties of large-scale preference training, beyond their initial focus on data efficiency gains; (2) we conduct preference fine-tuning on larger-scale and multiple preference datasets, and evaluate on rich benchmarks, expanding beyond their experiments which were limited to 5K samples and in-distribution testing due to resource constraints at the time.

## 3 MODELING WORLD PREFERENCE

### 3.1 EXPERIMENTAL SETUP

**Data Collection**: To model world preference, we collect data from multiple public forums, including StackExchange (a professional Q&A platform), Reddit (a social news and community discussion platform), and Quora (a knowledge sharing community). These forums contain posts with user responses that receive upvotes and downvotes, naturally creating preference patterns. For each prompt $x$ (post), we sample two responses with different net votes (upvotes minus downvotes) to form preference pairs $(y_0, y_1)$, where the label $Y$ points to the response with higher votes.

**Data Selection**: Through extensive training and validation across different data sources, Stack-Exchange emerges as the highest-quality source with superior out-of-domain generalization. We therefore select it as our primary forum data source for modeling world preference. Detailed analyses of different data sources can be found in Appendix A.1. Additional details on forum data analysis, preference pair sampling strategies (including vote margin controls) and training examples are provided in Appendices A.2, A.3 and G.

**Training Methods**: Our approach to world preference modeling follows the general human preference modeling framework as described in Section 2.1. We use a learning rate of 3e-6 and a batch size of 10K across all model sizes. Ablation studies on these settings are in Appendix B.1.

**Evaluation Methods**: We evaluate WorldPM using diverse test sets from multiple RM benchmarks spanning several subtasks: PPE (Frick et al., 2024) (Human, MMLU-Pro, IFEval, GPQA, MATH, and MBPP-Plus) , RMB (Zhou et al., 2024) (Helpfulness, Harmlessness) , RM-Bench (Liu et al., 2024) (chat, code, math, and safety) , RewardBench (Lambert et al., 2024) (chat, chat-hard, reasoning, and safety) , Offset Bias (Park et al., 2024), and HelpSteer2 (While HelpSteer2 will be used later for training, we include it here as a distinct source to test WorldPM's subjective performance) (Wang et al., 2024c). We uniformly use their preference pairs to compute test losses (i.e., BT loss) for consistent evaluation. Detailed descriptions of these benchmarks and our evaluation methods are provided in Appendices B.2 and B.3.

The capabilities tested by the above benchmarks can be broadly classified into three categories: (1) adversarial (identifying flaws in responses, such as constructing irrelevant rejected responses). (2) objective (identifying correct responses for queries with ground-truth answers), and (3) subjective (including human or AI subjective preferences).

### 3.2 SCALING TRENDS

We collect as many posts, responses, and votes as possible from StackExchange to construct preference pairs, yielding approximately 15M pairs and conduct world preference modeling on Qwen2.5 models (Yang et al., 2024b) ranging from 1.5B to 72B parameters. Subsequently, we evaluate WorldPM's out-of-domain generalization capabilities across various test sets using BT loss (Equation 2). We find that WorldPM exhibits different scaling properties across various domains and model sizes, as illustrated in Figure 1. Overall, WorldPM demonstrates scaling trends on test sets with well-defined answers (Objective and Adversarial), where more challenging test sets require larger models to achieve scaling benefits. However, no clear scaling trends are observed on test sets with open-ended answers (Subjective).

In Appendix C.2, we provide the task-wise scaling law fitting results in Figure 14, and additionally present views of how the loss varies with training compute (Figure 15).

**We observe a power law decrease in test losses in adversarial evaluation.** Our adversarial evaluation comprises three test sets: factual error detection (from RM-Bench chat), Offsetbias, and LLMBar (from RewardBench chat-hard). These test sets challenge model robustness through fac-

| Model | Adversarial | Objective |
|---|---|---|
| 1.5B | $L = 3.166D^{-0.103}$ (.991) | $L = 0.724D^{-0.003}$ (.420) |
| 7B | $L = 2.809D^{-0.111}$ (.995) | $L = 0.724D^{-0.008}$ (.652) |
| 32B | $L = 2.427D^{-0.103}$ (.979) | $L = 0.867D^{-0.021}$ (.943) |
| 72B | $L = 1.832D^{-0.090}$ (.937) | $L = 0.815D^{-0.018}$ (.944) |

Table 1: Power-law fitting results. Numbers in parentheses indicate $R^2$ values.

tual errors and off-topic/incomplete responses. Models of all sizes exhibit a consistent power-law decrease in test losses across these sets. We average the three test losses and fit the power-law

relationship between test loss ($L$) and training data size ($D$) as shown in Table 1, where the coefficient of determination ($R^2$) exceeds 0.9, indicating excellent scalability of human preference on such tasks. Prior studies (Park et al., 2024; Zeng et al., 2024) have identified PM vulnerabilities that RLHF can exploit (e.g., favoring superficially good but unfaithful responses). Our findings suggest these vulnerabilities may stem from limited training data, and PMs demonstrate increasingly stronger capability to identify flawed responses as training data expands.

**We observe an emergent phenomenon in objective metrics, where larger models demonstrate power law decrease in test losses across more benchmarks.** We evaluate multiple aspects using six objective test sets: coding (MBPP-Plus, HumanEvalPack), mathematics (MATH), knowledge-based QA (MMLU-Pro, GPQA), and instruction following (IFEval). MATH and HumanEvalPack are sourced from RM-Bench's code and math subsets, while others from PPE. Our analysis reveal a clear model scale effect: The 1.5B model fails to generalize on any test set. The 7B model shows weak scaling trends only on MBPP-Plus and IFEval, while the 32B model demonstrates further scaling on HumanEvalPack, MATH, and MMLU-Pro. In contrast, the 72B model exhibits consistent scaling trends across all test sets. We average the losses across these test sets and fit power laws for different model sizes (Table 1). We find that the $R^2$ values are extremely low for 1.5B and 7B models, while in contrast, 32B and 72B models show much better fitting results. These results indicate that preference modeling is an inherently challenging task, where **certain capabilities emerge only at larger model scales** - consistent with emergence phenomenon in language modeling (Wei et al., 2022).

**Test losses in subjective evaluations quickly converge.** We analyze annotations from three distinct sources: (1) **crowdsourced annotations** from PPE's Human Preference test set (collected from ChatBot Arena); (2) **expert annotations** from HelpSteer2; and (3) **AI annotations** (GPT4) from RMB's Helpfulness evaluation, using human-defined criteria to assess preferences. Models across all scales (1.5B to 72B) show no reduction in test losses with increased training data. While the 1.5B model performs distinctly worse, the 7B, 32B and 72B models are indistinguishable in subjective evaluations - contrary to patterns in other evaluations.

**We hypothesize that this might be due to conflicts between WorldPM and subjective evaluation in certain dimensions.** Subjective preference inherently involves multiple dimensions, such as helpfulness, relevance, and conciseness (Li et al., 2024). Individual preferences vary - some favor concise responses while others prefer detailed ones. Moreover, interpretations within the same dimension can differ across evaluators; one might find response A more helpful while others consider B more helpful. Such preference variations in test sets introduce evaluation uncertainty (Dubois et al., 2024; Park et al., 2024). A preference model optimized for brevity might appear to perform poorly when evaluated against annotations favoring verbosity, despite being effective in other dimensions like helpfulness. Examples of such ambiguous annotations from evaluation are shown in Appendix G.

## 3.3 WHY DON'T SUBJECTIVE EVALUATIONS SCALE? A STYLE-BASED ANALYSIS

### 3.3.1 STYLE-CONTENT SEPARATION IN EVALUATION

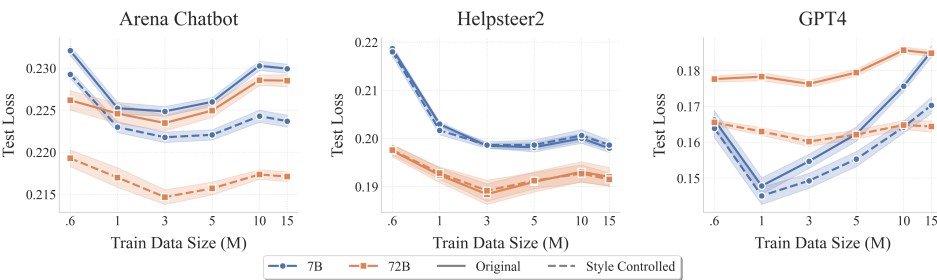

Figure 2: Comparison of test loss on subjective evaluation sets across annotation sources, with and without style control. HelpSteer2's expert annotations show minimal impact from style control, while crowdsourced annotations (ChatBot Arena) and AI annotations (GPT4) show substantial variations. The gap between controlled and uncontrolled conditions grows with training data and model size.

To investigate challenges in subjective evaluation, we focus on style preference as a reliably quantifiable aspect, which has been shown to influence LLM evaluation (Zheng et al., 2023a; Dubois et al., 2024; Chiang et al., 2024; Feuer et al., 2024).

Inspired by Chiang et al. (2024), we propose to separate style from content evaluation during the preference model assessment. For a preference pair $(x, y_0, y_1)$, let $r_\theta(x, y_0)$ and $r_\theta(x, y_1)$ be the model scores, with their *score difference* $D(x, y_0, y_1) = r_\theta(x, y_0) - r_\theta(x, y_1)$. Let $V \in \mathbb{R}^S$ be the style features vector, where $S$ is the feature count. We use four style features: token length, markdown lists, headers, and bold elements, consistent with Chiang et al. (2024). The *style difference* is normalized and scaled by the sum to ensure relative comparisons across different style features:

$$Z_i(x, y_0, y_1) = \text{normalize}\left(\frac{V_i(y_0) - V_i(y_1)}{V_i(y_0) + V_i(y_1)}\right), \quad i = 1, \ldots, S. \tag{3}$$

The final evaluation metric combines score difference $D$ and style difference $Z$ as $R = D\alpha + Z^T\beta$, where $\alpha$ and $\beta$ are weights for score and style differences. With the preference model and its scores frozen, the optimal weights are learned through linear regression by minimizing the binary cross-entropy loss:

$$\hat{\alpha}, \hat{\beta} = \arg\min_{\alpha \in \mathbb{R}, \beta \in \mathbb{R}^S} \frac{1}{n} \sum_{i=1}^{n} -(Y_i \log(\text{sigmoid}(R_i)) + (1 - Y_i) \log(1 - \text{sigmoid}(R_i))), \tag{4}$$

From an intuitive perspective, explicitly modeling stylistic factors during evaluation maximizes the stylistic gain in evaluation outcomes across different models, thus effectively eliminating stylistic influences when conducting comparisons across various models.

We compare test losses with and without style control across subjective test sets (style control uses $R_i = D_i\alpha + Z_i^T\beta$ as the final score, while the no-control setting fixes $\beta = 0$[1]). We randomly split each test set into two splits, fit the model on split A, and report the test loss on split B; this procedure is repeated five times to compute the standard deviation, as shown in Figure 2. The results on expert-annotated data (HelpSteer2) remain stable, likely due to the strict quality control in its construction. In contrast, test sets based on crowdsourced annotations (ChatBot Arena) and AI annotations (GPT-4) exhibit substantial variation. This suggests that **without careful annotation, subjective evaluations are highly sensitive to style factors.**

The gap between controlled and uncontrolled conditions **widens** with increased training and model scale, aligning with WorldPM's reduced style preference (discussed later). After style control, the 72B model surpasses the 7B model, though slightly underperforming on GPT4 evaluations - possibly due to additional uncontrolled preferences. This confirms the benefits of model scaling for subjective tasks, though the impact of training scale remains unclear - we suspect that world preferences may still conflict with certain preference in subjective evaluations.

Style analysis across other domains and feature ablation studies are provided in Appendix D. Although stylistic influence varies across different test sets, it does not alter the primary trends observed.

### 3.3.2 STYLE EFFECTS ON TRAINING DYNAMICS

Our previous analysis reveal style preference in subjective evaluation, showing increasing distortion in evaluation results as training scale and model size expand. To understand why this style bias amplifies, we examine style influence during WorldPM's training.

**We find that models initially over-rely on style features but reduce this bias with increased training data and model size.** To measure style influence on WorldPM predictions, given a preference pair $(x, y_0, y_1)$, we define $S(x, y_0, y_1)$ to indicate which response has better stylistic features, $P(x, y_0, y_1)$ represents the model's prediction of which response is preferable, and $G(x, y_0, y_1)$ is the human preference label. We use the Phi coefficient ($\phi$) to determine their correlation, which characterizes the correlation between two binary variables (Cramér, 1946). Specifically, $\phi(P, S)$ measures how strongly model predictions are influenced by stylistic features, while $\phi(G, S)$ reflects the relationship between human preferences and style.

---

[1]This differs from the original test loss calculation (where $\alpha = 1, \beta = 0$) because $\alpha$ is still optimized for a fair controlled comparison, which leads to a lower loss.

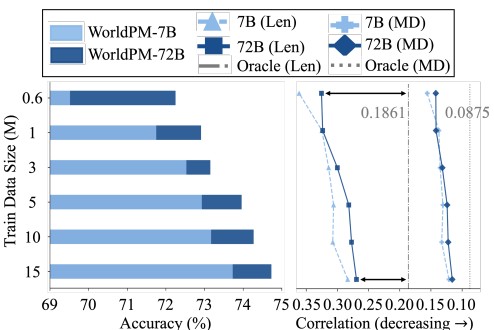 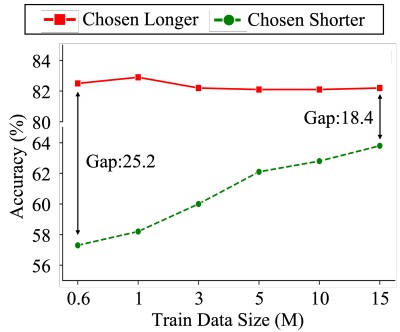

(a) The left panel shows continuous performance improvements, while the right demonstrates decreasing style-prediction correlation with larger models and more training. However, this correlation remains above the Oracle (style-label correlation), suggesting over-reliance on stylistic features.

(b) An asymmetric learning dynamic in style preference mitigation. The model rapidly fits majority-style (Chosen Longer) data while gradually incorporating minority-style instances (Chosen Shorter), maintaining performance on the former (Chosen Longer) throughout training.

As shown in Figure 3a, we conduct style preference analysis on the in-distribution validation set. The left panel demonstrates improving valid accuracy with increased training and model scale, indicating enhanced model performance. For analyzing stylistic influence, we consider two factors: $S_{length}$ (indicating which response is longer) and $S_{markdown}$ (indicating which response contains more markdown markers). The right panel shows that with larger training data and model sizes, the correlation between model predictions $P$ and these stylistic features $S$ tended to decline, suggesting the model's predictions became gradually less dependent on stylistic features.

The vertical lines represent the correlation between human preference labels $G$ and style $S$. While human preferences show positive correlation with stylistic features, this correlation is consistently lower than that between model predictions and style, which suggests that the model over-relies on stylistic features for predictions. This observation aligns with (Geirhos et al., 2020; Holtzman et al., 2021; Zhang et al., 2016): when training is insufficient or the model's capabilities are limited, it tends to prioritize surface-level, easily discernible features for making predictions.

Furthermore, we observe that both human labels and model predictions show stronger correlations with text length compared to markdown marker frequency, confirming previous findings(Chiang et al., 2024) that markdown usage serves as a secondary stylistic feature.

**The model quickly learns majority-style patterns while gradually adapting to minority-style cases.** We split the validation set into two length-based groups: Chosen Longer (chosen responses longer than rejected ones) and Chosen Shorter (vice versa). As shown in Figure 3b, we find that the model could classify Chosen Longer data with an 82% accuracy rate at a very early stage, indicating that it easily learns to distinguish this portion of the data. In contrast, the accuracy for Chosen Shorter data started at only 57%. As training progressed, the accuracy for Chosen Longer data remained stable overall, whereas the accuracy for Chosen Shorter data exhibited a trend of logarithmic growth with training volume. This reveals the general learning process of WorldPM: the model first captures preference patterns exhibited by the majority of the dataset. However, the remaining data demonstrates contrary characteristics, forcing the model to discover **underlying representations** that can simultaneously model both majority and minority cases to further reduce loss, thereby overcoming initial biases.

### 3.4 ALIGNMENT PERFORMANCE

To further understand WorldPM's subjective performance, we evaluate its preference modeling indirectly using two alignment benchmarks (Arena Hard (Li et al., 2024) and Alpaca Eval (Dubois et al., 2024)) that use AI judges to assess responses based on human preference criteria. Specifically, we use Best-of-N sampling to align language models: generating 256 responses from Qwen2.5-7B-Instruct for each benchmark query and letting different WorldPM checkpoints select the best one.

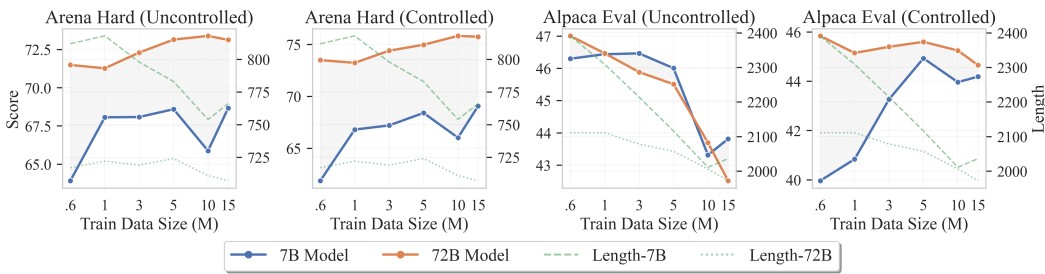

Figure 4: WorldPM alignment performance across training scales. Both benchmarks implement style control to reduce style bias in AI evaluation. Arena Hard shows stable trends with and without control: 72B consistently outperforms 7B and achieves optimal performance at larger training scales. In contrast, Alpaca Eval scores strongly correlate with style features without style control.

The results are shown in Figure 4, presenting both the original scores and scores after applying their respective style control strategies.

Selected response lengths decrease across both benchmarks, with 72B generating shorter responses than 7B, reflecting its reduced style dependency shown before. Arena Hard shows stable trends with and without style control, where 72B significantly outperforms 7B. For Alpaca Eval, performance strongly correlates with length without style control, making 7B and 72B indistinguishable; only with style control do clear differences emerge. The 72B model reaches optimal performance at different training scales: 0.6M for Alpaca Eval, 3M for direct RM evaluation (test loss above), and continues improving until 10M for Arena Hard.

Comparing evaluator prompts (Figure 11, 12), Arena Hard explicitly requires helpful, relevant, and concise (consistent with WorldPM's reduced style dependency) answers. In contrast, Alpaca Eval simply asks models to select the best output from a human perspective and lacks Arena Hard's Chain-of-Thought (COT) process. Consequently, without length control, Alpaca Eval results are strongly correlated with response length. This explains why Arena Hard performance continues improving with training while Alpaca Eval stops improving early.

This comparison highlights the challenges in subjective evaluation. **As training progresses, WorldPM may improve in aspects like conciseness, but some subjective evaluations (like Alpaca Eval) might prefer complex and detailed responses, which conflicts with the goal of conciseness**. Such divergence between WorldPM's scaling direction and evaluation criteria might exist in multiple other dimensions, leading to increasing test losses that appear as overfitting.

## 4 PREFERENCE FINE-TUNING BASED ON WORLDPM

### 4.1 EXPERIMENTAL SETUP

We evaluate WorldPM as an effective foundation for preference fine-tuning using three human preference datasets: HelpSteer2 (Wang et al., 2024c), UltraFeedback (Cui et al., 2023), and pair_data_v2 800K wsafety[2] from RLHFlow (hereafter referred to as RLHFlow). HelpSteer2 comprises approximately 7K preference comparisons across five dimensions. Since we consider only a single reward, we selected helpfulness scores as the preference labels. UltraFeedback contains four responses per question; we extracted approximately 100K preference pairs. RLHFlow encompasses approximately 800K data points. These datasets of varying sizes enable us to understand the effectiveness boundary of WorldPM. We use batch size 128 for HelpSteer2 and 512 for others, training each for at least two epochs. Models are selected based on validation loss.

We validate the effectiveness of WorldPM in preference fine-tuning through two approaches. First, we evaluate using various RM benchmark metrics (consistent with Section 3.1, including subjective, objective, and adversarial metrics). Following Section 3.3.1, we compute scores as preference prediction accuracy, where preference is determined by the sign of reward difference $R$ (with or

---

[2]https://huggingface.co/datasets/RLHFlow/pair_data_v2_80K_wsafety

without style control). Second, we employ best-of-N sampling for alignment evaluation. The alignment performance is assessed on two benchmarks - Arena Hard and Alpaca Eval - following our previously described evaluation protocol. Specifically, we generate 64 samples from Qwen2.5-7B-Instruct and Qwen2.5-72B-Instruct for the 7B and 72B models.

Table 2: Effect of WorldPM initialization on 72B model's PM fine-tuning performance across different evaluation categories. Abbreviations: RB - RewardBench, RM-B - RM-Bench.

| Metrics | Helpsteer2 | | UltraFeedback | | RLHFlow | |
|---|---|---|---|---|---|---|
| | w/o WorldPM | w/ WorldPM | w/o WorldPM | w/ WorldPM | w/o WorldPM | w/ WorldPM |
| **Subjective Evaluation** | | | | | | |
| PPE-Human | 64.76 | 66.68 (↑2.97%) | 65.12 | 66.94 (↑2.80%) | 68.14 | 68.93 (↑1.16%) |
| RMB-Helpful | 77.76 | 78.53 (↑1.00%) | 79.80 | 81.73 (↑2.41%) | 79.26 | 80.89 (↑2.05%) |
| RB-Chat | 98.04 | 98.88 (↑0.85%) | 97.63 | 96.37 (↓1.29%) | 98.46 | 97.91 (↓0.57%) |
| **Objective Evaluation** | | | | | | |
| PPE-MBPP | 70.50 | 78.76 (↑11.72%) | 75.02 | 79.46 (↑5.92%) | 82.28 | 82.39 (↑0.14%) |
| PPE-GPQA | 57.66 | 59.63 (↑3.42%) | 59.22 | 61.19 (↑3.33%) | 62.52 | 63.75 (↑1.97%) |
| PPE-IFEval | 59.24 | 63.63 (↑7.42%) | 60.57 | 64.34 (↑6.22%) | 64.26 | 65.39 (↑1.76%) |
| PPE-MATH | 69.94 | 77.17 (↑10.33%) | 72.77 | 78.16 (↑7.41%) | 79.36 | 83.01 (↑4.60%) |
| PPE-MMLU | 70.20 | 74.96 (↑6.79%) | 72.42 | 75.08 (↑3.67%) | 79.04 | 79.30 (↑0.32%) |
| RM-B-Code | 60.75 | 69.30 (↑14.08%) | 67.32 | 70.18 (↑4.23%) | 72.15 | 73.46 (↑1.82%) |
| RM-B-Math | 72.78 | 77.69 (↑6.75%) | 73.72 | 77.79 (↑5.51%) | 87.81 | 86.20 (↓1.83%) |
| RB-Reason | 96.65 | 98.25 (↑1.66%) | 96.30 | 97.48 (↑1.23%) | 97.52 | 98.15 (↑0.64%) |
| **Adversarial Evaluation** | | | | | | |
| RB-Chat-Hard | 84.87 | 87.28 (↑2.84%) | 84.54 | 83.00 (↓1.82%) | 84.21 | 84.54 (↑0.39%) |
| RM-B-Chat | 81.01 | 82.17 (↑1.44%) | 80.23 | 79.46 (↓0.97%) | 85.66 | 84.50 (↓1.36%) |
| OffsetBias | 83.14 | 89.76 (↑7.96%) | 87.29 | 88.05 (↑0.88%) | 87.89 | 87.47 (↓0.47%) |
| **Alignment Evaluation** | | | | | | |
| Arena Hard | 88.52 | 89.72 (↑1.36%) | 89.16 | 90.37 (↑1.36%) | 90.41 | 90.82 (↑0.45%) |
| Alpaca Eval | 60.34 | 63.39 (↑5.07%) | 59.95 | 63.73 (↑6.31%) | 65.51 | 66.26 (↑1.14%) |
| Arena Hard-SC | 88.47 | 90.54 (↑2.34%) | 88.82 | 90.74 (↑2.16%) | 91.06 | 91.76 (↑0.77%) |
| Alpaca Eval-LC | 53.26 | 55.45 (↑4.12%) | 56.83 | 59.80 (↑5.23%) | 56.95 | 55.58 (↓2.40%) |

## 4.2 EVALUATION RESULTS

Table 2 shows the results for 72B models with style-controlled scores (see Appendix C.4 for 7B and uncontrolled results, which show consistent findings). **WorldPM initialization improves performance across domains and fine-tuning datasets of varying scales.** Key findings include:

**Subjective domains**: PPE-Human and RMB-Helpful show notable improvements, suggesting WorldPM learns useful general representations for subjective preference. RewardBench-Chat shows slight decrease but remains above 97%, indicating metric saturation.

**Objective domains**: PPE's objective metrics, RM-Bench's code and math metrics, and Reward-Bench's reasoning metrics demonstrate broad improvements.

**Adversarial domains**: Smaller datasets (e.g., HelpSteer2) show notable improvements, while larger ones exhibit minimal changes. WorldPM itself already achieves high accuracy on these metrics (around 90% for RM-Bench Chat and OffsetBias, as shown in Figure 19). The post-fine-tuning decrease shows WorldPM's initial gains are gradually lost, approaching the performance of models without WorldPM initialization, indicating easily modifiable features.

**Alignment evaluation**: Most metrics improve with and without style control (-LC and -SC denote their respective default style control strategies). Exceptions in RLHFlow's Alpaca Eval may be attributed to RLHFlow's large scale and high quality, making WorldPM's benefits less pronounced.

Overall, preference fine-tuning of different scales benefit from WorldPM initialization. Given the high cost of preference annotation, WorldPM provides a valuable foundation for PM training.

## 4.3 THE IMPACT OF WORLDPM TRAINING SCALE ON FINE-TUNING

To study how WorldPM's training scale affects fine-tuning, we experiment with different 72B WorldPM checkpoints (trained on 5M, 10M and 15M samples) and a baseline without WorldPM. We fine-tune each checkpoint on Helpsteer2 and Ultrafeedback datasets, selecting models with the lowest validation loss. Our evaluation encompassed style-controlled PPE metrics (averaging all objective metrics) and downstream alignment performance assessed through BoN sampling on Qwen2.5-72B-Instruct using AlpacaEval and Arena-Hard benchmarks.

As shown in Figure 5, we observe significant improvements starting at 5M samples, aligning with scaling laws where linear gains require exponential data growth. PPE-Object shows the most consistent improvements with increased data, while subjective metrics (all of other metrics) show more variation. Overall, we can observe that **larger-scale WorldPM achieves better performance on fine-tuning.**

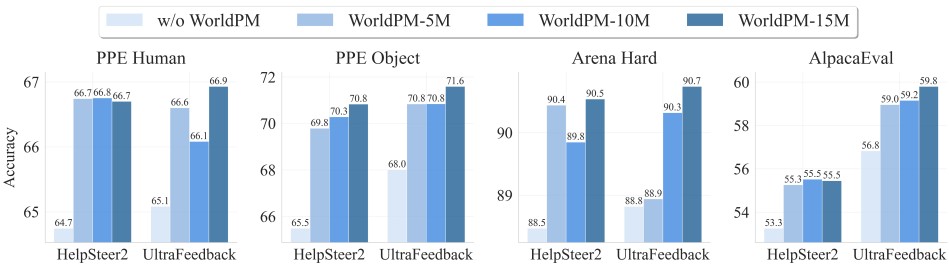

Figure 5: Comparison of PM fine-tuning performance across different WorldPM training scales and baseline without WorldPM. Larger WorldPM scales demonstrate enhanced fine-tuning benefits.

## 5 CONCLUSION

Our work reveals that preference modeling follows Scaling Laws similar to language modeling. We validate this on objective and adversarial tests, while analyzing why subjective evaluations don't scale similarly. We also demonstrate WorldPM's effectiveness for preference fine-tuning. This study highlights both the potential and challenges in preference modeling, calling for new benchmarks to reveal this potential and improved frameworks to enhance scaling efficiency.

## ETHICS STATEMENT

**Purpose and Intended Use**

WorldPM is strictly intended for academic research purposes, specifically for: (1) investigating the scaling laws of preference modeling (reward modeling), and (2) conducting preference fine-tuning experiments to enhance fine-tuning effectiveness based on the WorldPM model.

**Potential Risks and Safeguards**

We acknowledge the potential risks associated with using large-scale forum data, including dual-use concerns, privacy issues, and group/cultural biases. To mitigate these risks:

1. We implement rigorous data anonymization during collection, gathering only essential fields (questions, responses, and vote counts) while removing links and personal information.

2. The model is strictly limited to non-generative purposes (reward modeling), preventing direct generation of potentially copyright-infringing or privacy-compromising content.

3. We strive to collect and balance data from diverse boards and demographics to minimize bias towards specific groups or cultures.

4. We utilize both the forum's moderation system and response voting metrics to ensure the quality and appropriateness of the training data. Specifically, content that has been flagged or removed by the forum's automated or human moderation processes is excluded

from our dataset. Additionally, we filter out responses that have received a significant number of downvotes, as these are often indicative of content that is misleading, low-quality, inappropriate, or otherwise deemed unhelpful by the community.

5. We will restrict WorldPM to academic research purposes and applications that benefit human welfare. To mitigate potential biases, we recommend fine-tuning the model based on WorldPM before deployment.

Despite our efforts to mitigate biases by collecting data from diverse sections and user groups, we acknowledge that WorldPM may exhibit inherent biases because its StackExchange training data is mostly in English and focused on technical topics. While we have identified and mitigated certain stylistic biases (such as length preferences) through our evaluation methodology, we recognize that unidentified biases may persist. For example, the model may favor technical or professional answers over creative ones, and its focus on English means it may not perform as well on other languages.

**Data Licensing and Usage**

This research incorporates data from three public platforms: StackExchange, Reddit, and Quora. While Reddit and Quora data were only used for data source selection analysis in the Appendix A.1, StackExchange serves as the sole data source for model training and analysis. The model we plan to release will be trained exclusively on StackExchange data.

1. StackExchange data is publicly available through data dumps [3] and is licensed under the CC BY-SA 4.0 license. We strictly adhere to this license and will release our model under the same license as required.

2. Reddit provides public access to their official API [4], through which we obtain their forum data. We fully adhered to Reddit's API Terms of Service.

3. We parse Quora corpus from Common Crawl [5]. Given Quora's Terms of Service, we will not release any Quora-related datasets or models. The Quora data used in our research has been anonymized and is limited to data source comparison purposes only.

## REPRODUCIBILITY STATEMENT

Algorithmically, WorldPM follows the same training approach as traditional BT models, with detailed training and evaluation settings provided in Appendix B. The training data for WorldPM is sourced from StackExchange, with data acquisition methods and quality enhancement through filtering mechanisms described in Section 5. Sections 3.1 and A provide comprehensive explanations of our data collection process and detailed data analysis. To ensure reproducibility, we fully disclose all details regarding training, evaluation, and data.

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

# A DETAIL OF DATA

## A.1 TRAINING DATA SELECTION

Table 3: Performance comparison of general human preference across different data sources. Stack-Exchange significantly outperforms other sources, approaching or surpassing open-source preference models (ArmoRM-Llama3-8B-v0.1). Math SE, a mathematics-specific board of StackExchange, demonstrates strong general human preference despite its domain-specific nature.

| Dataset | PPE-Human | PPE-Objective | RMB | RewardBench | RM-Bench |
|---|---|---|---|---|---|
| Quora | 60.2 | 57.7 | 66.1 | 69.0 | 60.9 |
| Reddit | 55.1 | 59.4 | 59.5 | 77.0 | 60.7 |
| StackExchange | 62.8 | **62.1** | **76.7** | **84.4** | 72.5 |
| ↪ *Math SE,* | **62.9** | 62.0 | 75.0 | 83.3 | **75.0** |
| ArmoRM | 60.2 | 64.3 | 72.3 | 89.8 | 75.4 |

We collect 800K preference pairs from each forum and train them on Qwen2.5-7B. As shown in Table 3, we evaluate them using multiple benchmarks, where PPE-Object is the average of five PPE objective evaluation sets. RMB, RewardBench, and RM-Bench calculate the average of their respective subsets. We also include ArmoRM-Llama3-8B-v0.1(Wang et al., 2024a) as a reference. It can be found that **StackExchange preference data shows the highest quality and demonstrates strong out-of-domain generalization ability, approaching or even surpassing open-source preference models.** Specifically, StackExchange data significantly outperforms Reddit and Quora across all evaluation sets, demonstrating its high-quality. Comparing StackExchange with ArmoRM, we find comparable performance on PPE-Objective and RM-Bench, while even surpassing ArmoRM on PPE-Human and RMB. Given that StackExchange's content is entirely out-of-domain from downstream general preference evaluations, its strong generalization ability is particularly impressive.

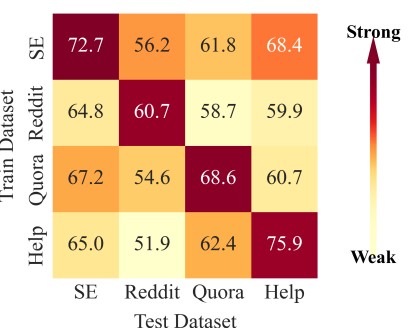

Figure 6: Cross generalization across different data sources including StackExchange, Reddit, Quora, and HelpSteer2, where models trained on one source and predict preferences on the others. The values indicate test accuracy.

To further understand StackExchange's out-of-domain generalization ability, we consider an extreme data source: StackExchange's Math board, which contains only mathematics-related content. Models trained on this board perform similarly to those trained on StackExchange data across various human preference benchmarks, indicating that **general human preferences can transfer across different domains.**

Besides, we verify StackExchange's generalization ability across different data sources, including HelpSteer2 as a few-human-annotated data source. Models are trained separately on StackExchange, Reddit, Quora, and HelpSteer2, then cross-evaluated on each other's data, as shown in the Figure 6, colors are normalized per column due to varying dataset difficulties, while values show original accuracies. **StackExchange shows the best generalization ability among multiple forum data sources and can represent different data sources.**

Given StackExchange's superior quality compared to other sources, we select it as the representative forum data source for modeling world preference.

## A.2 FORUM DATA ANALYSIS

**Different dimensions of forum data generally follow long-tail distributions**. We analyze all the data collected from StackExchange, as shown in Figure 7. Our analysis focuses on three key dimensions: reply length, reply upvotes, and the number of replies per post. We find that forum data, or naturally generated human data, typically exhibits strong long-tail characteristics. For instance,

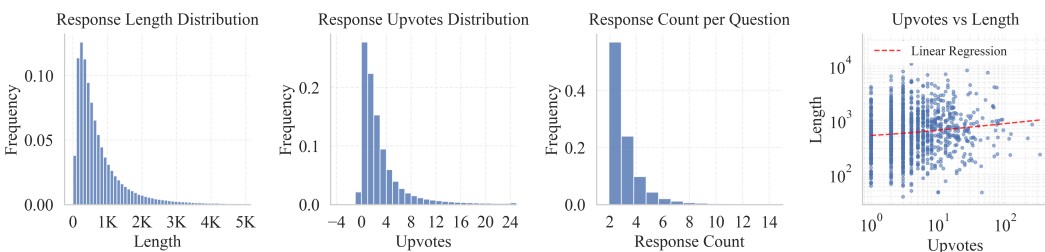

Figure 7: Distribution of reply lengths, upvotes, and the number of replies per post across all StackExchange forum data. All these characteristics demonstrate evident long-tail distributions. The last subplot indicates a weak correlation between reply length and upvotes.

most forum replies are under 1K in length, receive fewer than 10 upvotes, and posts typically have fewer than 6 replies. Notably, the majority of posts have only one reply, and many replies have zero upvotes. Consequently, we need to filter out a significant portion of data where preference relationships cannot be defined, as establishing a preference pair requires at least two replies to a post with different numbers of upvotes.

It's worth noting that there are relatively few extremely short replies in the forum data, as shown in the first graph. We hypothesize this is due to StackExchange's relatively strict content moderation mechanisms, which filter out replies lacking substantial information.

We further analyze the relationship between reply length and upvotes, as shown in the last column of the figure. Overall, the distribution of reply lengths appears relatively uniform across different upvote counts, although there remains a positive correlation between reply length and upvotes. This correlation is natural in human-generated content, as longer replies typically indicate greater effort from the author, potentially resulting in higher quality content.

### A.3 PREFERENCE DATA CONSTRUCTION

On StackExchange, users can upvote or downvote replies, and post authors can choose to accept specific replies. We incorporate the author's acceptance as an additional upvote. The final score for each reply is calculated by subtracting downvotes from upvotes. Using these scores, we define preference directions: when two replies have different scores, we consider them to form a preference relationship.

**Score differences do not affect preference performance.** One potential concern is whether replies with close scores (e.g., one reply with 1 upvote versus another with 2 upvotes) lack sufficient distinction. To address this, we bucket the score differences between reply pairs into groups: 1-2, 3-5, 6-10, and 11+ differences. As shown in Table 4, we find similar performance across different benchmarks for each bucket, with no bucket showing significant performance variations.

Table 4: Bucketing preference pairs by score differences to evaluate the impact of score gaps on preference data quality.

| Score Diff | PPE-Human | PPE-Objective | RMB | RewardBench | RM-Bench |
|---|---|---|---|---|---|
| 1-2 | **63.1** | 59.4 | 73.8 | 79.9 | **75.2** |
| 3-5 | 62.6 | 59.5 | **76.7** | 84.2 | 73.6 |
| 6-10 | 62.3 | 59.3 | 76.1 | **85.5** | 73.2 |
| $\geq 11$ | 62.5 | **60.5** | 74.2 | 85.2 | 71.1 |

**Similar performance across different topic domains.** Another potential concern is that different boards of StackExchange, covering diverse content areas, might affect downstream generalization, especially given that our downstream evaluation spans general dialogue, mathematics, coding, and other aspects. To address this, we divide StackExchange data into three segments: StackOverflow (computer science-related topics), Math StackExchange (mathematics-related topics), and Others.

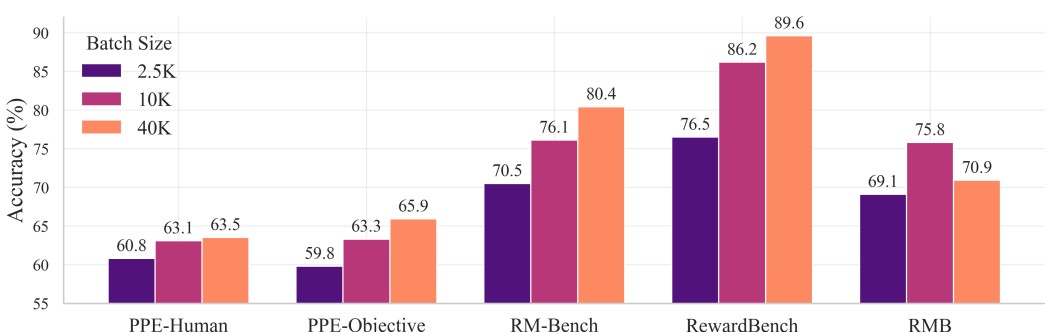

Figure 8: Comparison of different batch sizes (from 2.5K to 40K) on Qwen2.5 7B with fixed 256 steps. Larger batch sizes consistently yield better performance.

This division reflects that the majority of StackExchange content comes from the first two sections, while the remaining 170+ sections contain diverse topics with relatively few questions each.

We train models separately on each section and evaluate them across all test sets, as shown in Table 5. Our results indicate no significant performance variations across different sections. Furthermore, StackExchange provides topic tags for each question, allowing for more granular topic identification. For instance, StackOverflow uses specific tags like c++, java, and pointers. These tags also follow a long-tail distribution, with common programming questions constituting the majority. We conduct additional experiments with topic resampling (StackOverflow-TR) on StackOverflow data, deliberately oversampling from less common tags to enhance data diversity. As shown in the table, this resampling approach yields similar results.

These findings suggest that human preferences are domain-agnostic: despite varying discussion topics, users within the same forum community demonstrate consistent preference patterns.

Table 5: Analysis of performance across different forum categories and topic distributions within individual categories indicates that human preferences exhibit robust cross-domain transferability.

| Category | PPE-Human | PPE-Objective | RMB | RewardBench | RM-Bench |
|---|---|---|---|---|---|
| StackExchange | 62.8 | 62.1 | 76.7 | 84.4 | 72.5 |
| Other | 62.7 | 62.0 | 76.2 | 85.7 | 72.3 |
| Math StackExchange | 62.9 | 62.0 | 75.0 | 83.3 | 75.0 |
| StackOverflow | 63.3 | 63.0 | 75.0 | 83.3 | 74.9 |
| StackOverflow-TR | 62.4 | 62.8 | 75.3 | 84.8 | 73.6 |

## B  DETAIL OF SETTINGS

### B.1  EXPERIMENTAL SETTINGS

We conduct world preference modeling experiments on Qwen2.5 models ranging from 1.5B to 72B parameters, with a batch size of 10K, training steps of 1536, and a learning rate of 3e-6. For smaller datasets and comparative experiments, we typically set the batch size to 2048. All experiments are performed on a cluster of 128×A100 GPUs. We use the Adam optimizer and employ learning rate warmup with a ratio of 0.1 and the weight decay coefficient of 0.1. We employ learning rate warmup with a ratio of 0.1, followed by a constant learning rate. The context length is set to 2048.

We discover that when modeling preferences, larger batch sizes lead to better performance under the same number of training steps and dataset size, as shown in Figure 8 and Figure 9. Performance continues to improve even with batch sizes up to 40K, indicating that preference modeling is indeed a challenging task that benefits from more accurate loss estimation through larger batch sizes. However, considering the limited training data available, we ultimately adopt a batch size of 10K to minimize loss noise while maintaining computational efficiency.

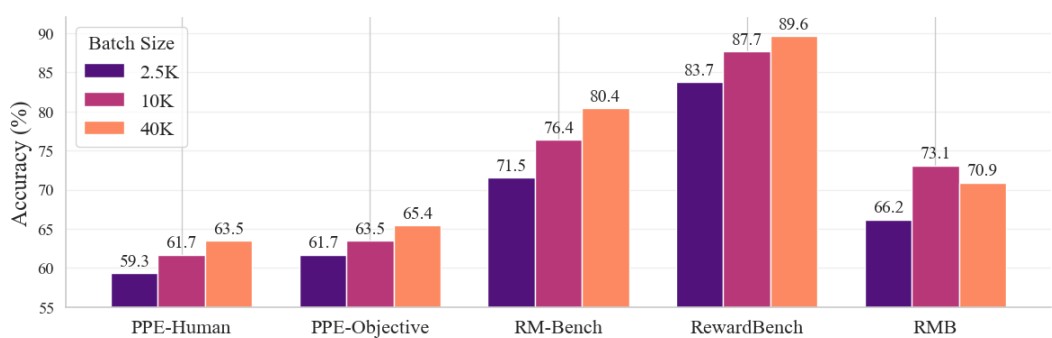

Figure 9: Comparison of different batch sizes (from 2.5K to 40K) on Qwen2.5 7B with fixed dataset size (10M). Larger batch sizes consistently yield better performance.

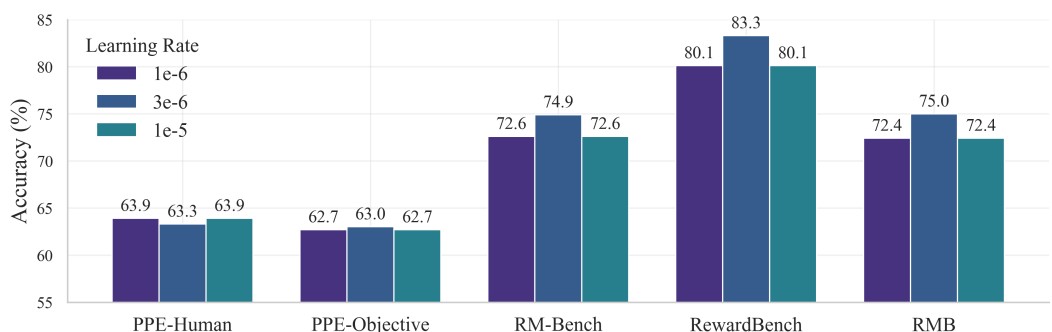

Figure 10: Through our experiments with different learning rates on Qwen2.5 7B, we observe that the model's final performance remains relatively stable across various learning rates, with 3e-6 showing slightly superior results. This suggests that the model's performance is robust to learning rate variations within a reasonable range.

We additionally conduct comprehensive ablation studies on learning rates, as shown in Figure 10. Our systematic comparison of learning rates at 1e-6, 3e-6, and 1e-5 demonstrates that model performance remains stable across this range, with 3e-6 exhibiting marginally superior results. This finding suggests that preference modeling is robust to learning rate variations within reasonable bounds. However, larger learning rates require more all-compute time, which is not practical for large-scale training.

## B.2 EVALUATION METHODS

To comprehensively evaluate WorldPM, we utilize different domain test sets from multiple RM benchmarks. Given our specific evaluation needs, we do not strictly follow the evaluation protocols provided with these benchmarks. Detailed explanations can be found in Appendix B.3. The benchmarks used in this work are as follows:

- **PPE** (Frick et al., 2024): This includes evaluations of both subjective and objective parts. The data for the subjective part comes from real user annotations in the Chatbot Arena[6]. The objective part collects queries from the MMLU-Pro (Wang et al., 2024b), IFEval (Zhou et al., 2023), GPQA (Rein et al., 2023), MATH (Hendrycks et al., 2021), and MBPP-Plus (Austin et al., 2021) datasets, and gathers responses from state-of-the-art models. The correctness of the responses is validated against real answers, thereby forming preference pairs (with *chosen* as correct and **rejected** as incorrect).

---

[6]https://lmarena.ai/

- **RMB** (Zhou et al., 2024): This benchmark relies on GPT4 as the primary annotator, with additional human verification processes. It encompasses diverse scenarios and is fundamentally divided into two aspects: helpfulness and harmlessness.

- **RM-Bench** (Liu et al., 2024): This consists of evaluations in four domains: chat, code, math, and safety. The chat domain evaluation assesses models' ability to identify factual errors by inserting them into responses. Prompts for code and math are sourced from the HumanEval-Pack (Muennighoff et al., 2023) and MATH (Hendrycks et al., 2021) datasets, respectively, with validation against real answers. The safety section includes both pseudo-harmful and genuinely harmful questions to evaluate the model's safety assessment capability.

- **Reward Bench** (Lambert et al., 2024): This includes evaluations in four domains: chat, chat-hard, reasoning, and safety. The chat-hard section, primarily sourced from LLMBar (Zeng et al., 2024), challenges reward models through the construction of subtly flawed responses designed to mislead evaluation.

- **Offset Bias** (Park et al., 2024): This dataset constructs high-quality but incorrect responses (rejected responses) to challenge reward models, including off-topic responses and responses containing errors.

- **HelpSteer2** (Wang et al., 2024c): This dataset, carefully annotated and filtered by trained human annotators, serves dual purposes: while it will be used as a training set in later sections, during this phase it functions as a test set for evaluating WorldPM's subjective performance.

## B.3 EVALUATION SETTINGS

We evaluate our preference models using multiple RM benchmarks, including PPE, RMB, RM-Bench, and RewardBench. However, we modify some of their evaluation methodologies for our specific context.

RMB offers PairWise and Best-of-N (BoN) evaluations, where BoN measures a model's ability to select the best reply from multiple options, and PairWise is essentially BoN with N=2. We focus on PairWise evaluation because: (1) BT loss computation inherently involves pairs of replies, making BoN evaluation computationally challenging, and (2) the original paper shows strong correlation between PairWise and BoN results.

Table 6: Correlation coefficients between response length and answer quality across different models.

| Model | Correlation |
|---|---|
| Gemma-2-9B-IT | $+0.047$ |
| GPT-4o-Mini | $-0.133$ |
| Llama-3-8B-Instruct | $-0.115$ |
| Claude-3-Haiku | $-0.619$ |

For RM-Bench, we utilize only the model's original responses to form preference pairs, rather than their style-modified versions, as we propose our own method for style-content separation evaluation in this paper.

For PPE, we identify a significant bias issue in the MBPP-Plus test set. This test set is constructed by sampling from four advanced models, where each model generates both correct and incorrect answers to form preference pairs. In our analysis, we observe unusually volatile metrics and conduct a detailed investigation. We define two binary variables: one indicating which answer is correct, and another indicating which answer is longer.

By measuring the correlation (Phi coefficient) between answer correctness and length, we find that Claude-3-Haiku samples show a strong negative correlation (-0.6) between these variables, where shorter answers are predominantly correct. This correlation significantly deviates from other models' patterns, indicating a severe style bias. Consequently, we exclude Claude-3-Haiku samples from MBPP-Plus test set to ensure reliable evaluation results.

## B.4 EVALUATOR PROMPTS

The evaluation prompts used in Arena Hard and Alpaca Eval are shown in Figure 11 and Figure 12, respectively.

**Arena Hard's Prompt**

```
Please act as an impartial judge and evaluate the quality of the
    responses provided by two AI assistants to the user prompt
    displayed below. You will be given assistant A's answer and
    assistant B's answer. Your job is to evaluate which assistant's
    answer is better.

Begin your evaluation by generating your own answer to the prompt.
    You must provide your answers before judging any answers.

When evaluating the assistants' answers, compare both assistants'
    answers with your answer. You must identify and correct any
    mistakes or inaccurate information.

Then consider if the assistant's answers are helpful, relevant, and
     concise. Helpful means the answer correctly responds to the
    prompt or follows the instructions. Note when user prompt has
    any ambiguity or more than one interpretation, it is more
    helpful and appropriate to ask for clarifications or more
    information from the user than providing an answer based on
    assumptions. Relevant means all parts of the response closely
    connect or are appropriate to what is being asked. Concise means
     the response is clear and not verbose or excessive.

Then consider the creativity and novelty of the assistant's answers
     when needed. Finally, identify any missing important
    information in the assistants' answers that would be beneficial
    to include when responding to the user prompt.

After providing your explanation, you must output only one of the
    following choices as your final verdict with a label:

1. Assistant A is significantly better: [[A>>B]]
2. Assistant A is slightly better: [[A>B]]
3. Tie, relatively the same: [[A=B]]
4. Assistant B is slightly better: [[B>A]]
5. Assistant B is significantly better: [[B>>A]]

Example output: "My final verdict is tie: [[A=B]]".
```

Figure 11: Evaluation prompts used in Arena Hard.

---

**Alpaca Eval's Prompt**

```
<|im_start|>system
You are a highly efficient assistant, who evaluates and selects the
    best large language model (LLMs) based on the quality of their
    responses to a given instruction. This process will be used to
    create a leaderboard reflecting the most accurate and human-
    preferred answers.
<|im_end|>
<|im_start|>user
I require a leaderboard for various large language models. I'll
    provide you with prompts given to these models and their
    corresponding outputs. Your task is to assess these responses,
    and select the model that produces the best output from a human
    perspective.

## Instruction

{
    "instruction": """{instruction}""",
}

## Model Outputs

Here are the unordered outputs from the models. Each output is
    associated with a specific model, identified by a unique model
    identifier.
{
    {
        "model_identifier": "m",
        "output": """{output_1}"""
    },
    {
        "model_identifier": "M",
        "output": """{output_2}"""
    }
}
## Task

Evaluate the models based on the quality and relevance of their
    outputs, and select the model that generated the best output.
    Answer by first providing a concise explanation and then end
    your answer by providing the model identifier of the best output
    . We will use the last character of your output 'output[-1]' as
    the name of the best model, so make sure you finish with the
    token of the model identifiers and nothing else: 'm' or 'M' (no
    quotes, no dots, no backticks, no new lines, ...). For example:

### Concise explanation
...some text...
### Which is best, m or M?
M

Now is your turn.

## Your answer: "Concise explanation" followed by "Which is best, m
    or M?"
<|im_end|>
```

Figure 12: Evaluation prompts used in Alpaca Eval.

# C ADDITIONAL EXPERIMENTAL RESULTS

## C.1 MOMENT OF EPIPHANY

During large-scale WorldPM training, we observe a **moment of epiphany**. As shown in Figure 13, the training loss exhibits logarithmic decrease with increasing training data volume. All training is completed **within a single epoch**, without reusing any training samples. At approximately 12.6M training samples, we observe a critical transition: a sudden drop in loss coincides with a sharp spike in gradients that quickly subsides. This synchronized pattern—the abrupt gradient surge indicating the discovery of a steep optimization direction, and the subsequent loss reduction suggesting a successful transition to a better solution space—implies that the model has discovered a more general and effective preference representation, potentially leading to stronger generalization capabilities across different preference scenarios.

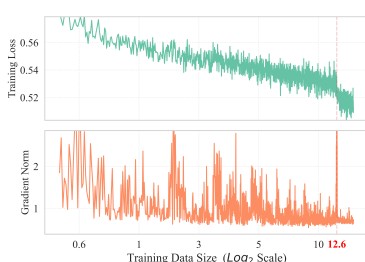

Figure 13: A moment of epiphany occurs during WorldPM training on Qwen 72B, characterized by a sudden drop in loss and a corresponding spike in gradients.

## C.2 FURTHER DETAILS ON THE SCALING LAW

For each task, we fit a power-law relationship of the form $L = \alpha D^\beta$ and report the corresponding $R^2$ values (Figure 14).
For objective tasks, more tasks exhibit a high goodness-of-fit as model size increases, indicating that their performance scaling can be well described by a simple power law.

We further present compute-normalized views (Figure 15), where we compare scaling trends under matched computational budgets. Overall, allocating more computation tends to yield better performance. Notably, for some tasks (e.g., GPQA, IFEval), a sufficiently large base model is required to effectively leverage additional computation; smaller models do not benefit as much from the same increase in compute. This observation is consistent with, and further supports, our findings regarding the emergence phenomena on objective tasks.

## C.3 SCALING TRENDS ACROSS ALL TEST SETS

Figure 16 illustrates the comprehensive scaling trends of world preference across all evaluation benchmarks. In HelpSteer2's five-dimensional assessment framework, we observe strong correlations among Helpfulness, Correctness, and Coherence dimensions, while Complexity and Verbosity demonstrate high mutual correlation. The latter two metrics show strong association with response length, as discussed in Section D.2. As elaborated in the main text, WorldPM demonstrates progressive mitigation of length preference, manifesting as increasing loss on these metrics.

Safety evaluation, omitted from the main text due to space constraints, encompasses RMB-Harmlessness, RewardBench-Safety, and RM-Bench-Safety metrics. RewardBench-Safety exhibits distinct scaling patterns across model sizes: the 1.5B model shows continuous decline, while 7B and 72B models stabilize post-decline, achieving accuracy exceeding 90%, indicating performance saturation. RMB-Harmlessness presents heterogeneous patterns: declining for 1.5B, saturating for 7B, and ascending for 72B. These diverse patterns can be partially interpreted through RM-Bench-Safety's subdomain analysis: pseudo-harmful queries versus genuinely harmful queries (RM-Bench-Safety-Response), as RMB-Harmlessness also incorporates both aspects. Our analysis reveals decreasing and saturating loss patterns for genuine harmful content detection, while pseudo-harmful query response shows ascending loss in later training stages. This suggests sustained efficacy in identifying genuinely harmful content, coupled with increased conservatism toward pseudo-harmful queries. We hypothesize that seemingly benign queries in specific contexts (e.g., gaming-related weapon discussions) might inadvertently enable harmful applications (e.g., transferable knowledge to real weapons).

| Task | 1.5B: $L = \alpha D^\beta$ ($R^2$) | 7B: $L = \alpha D^\beta$ ($R^2$) | 32B: $L = \alpha D^\beta$ ($R^2$) | 72B: $L = \alpha D^\beta$ ($R^2$) |
|---|---|---|---|---|
| Factual Error Detect | $L = -0.273D^{-0.129}$ (0.998) | $L = -0.404D^{-0.148}$ (0.991) | $L = -0.515D^{-0.106}$ (0.983) | $L = -0.600D^{-0.117}$ (0.974) |
| OffsetBias | $L = -0.308D^{-0.129}$ (0.991) | $L = -0.701D^{-0.093}$ (0.975) | $L = -0.664D^{-0.132}$ (0.986) | $L = -0.773D^{-0.091}$ (0.902) |
| LLMBar | $L = -0.239D^{-0.061}$ (0.882) | $L = -0.417D^{-0.086}$ (0.944) | $L = -0.449D^{-0.073}$ (0.919) | $L = -0.548D^{-0.056}$ (0.842) |
| MBPP-Plus | $L = -0.373D^{-0.005}$ (0.673) | $L = -0.491D^{-0.022}$ (0.711) | $L = -0.496D^{-0.040}$ (0.911) | $L = -0.507D^{-0.021}$ (0.903) |
| MMLU-Pro | $L = -0.371D^{-0.001}$ (0.027) | $L = -0.449D^{-0.017}$ (0.715) | $L = -0.466D^{-0.038}$ (0.957) | $L = -0.513D^{-0.033}$ (0.897) |
| IFEval | $L = -0.349D^{-0.001}$ (0.113) | $L = -0.367D^{-0.003}$ (0.633) | $L = -0.369D^{-0.004}$ (0.891) | $L = -0.384D^{-0.011}$ (0.957) |
| HumanEvalPack | $L = -0.362D^{-0.005}$ (0.493) | $L = -0.390D^{0.003}$ (0.310) | $L = -0.400D^{-0.017}$ (0.968) | $L = -0.419D^{-0.019}$ (0.959) |
| GPQA | $L = -0.372D^{0.000}$ (0.014) | $L = -0.370D^{0.004}$ (0.706) | $L = -0.380D^{-0.002}$ (0.208) | $L = -0.384D^{-0.007}$ (0.962) |
| MATH | $L = -0.365D^{-0.005}$ (0.389) | $L = -0.424D^{-0.041}$ (0.923) | $L = -0.492D^{-0.056}$ (0.966) | $L = -0.469D^{-0.033}$ (0.901) |
| Chatbot Arena | $L = -0.387D^{0.009}$ (0.803) | $L = -0.435D^{0.002}$ (0.068) | $L = -0.425D^{-0.003}$ (0.462) | $L = -0.441D^{0.004}$ (0.400) |
| HelpSteer2 | $L = -0.388D^{0.002}$ (0.153) | $L = -0.493D^{-0.011}$ (0.453) | $L = -0.475D^{-0.013}$ (0.748) | $L = -0.529D^{-0.005}$ (0.224) |
| GPT4 | $L = -0.464D^{0.027}$ (0.709) | $L = -0.638D^{0.009}$ (0.066) | $L = -0.586D^{-0.004}$ (0.149) | $L = -0.594D^{0.002}$ (0.018) |

Figure 14: Power-law fits of the form $L = \alpha D^\beta$ for each task, along with the corresponding $R^2$ values.

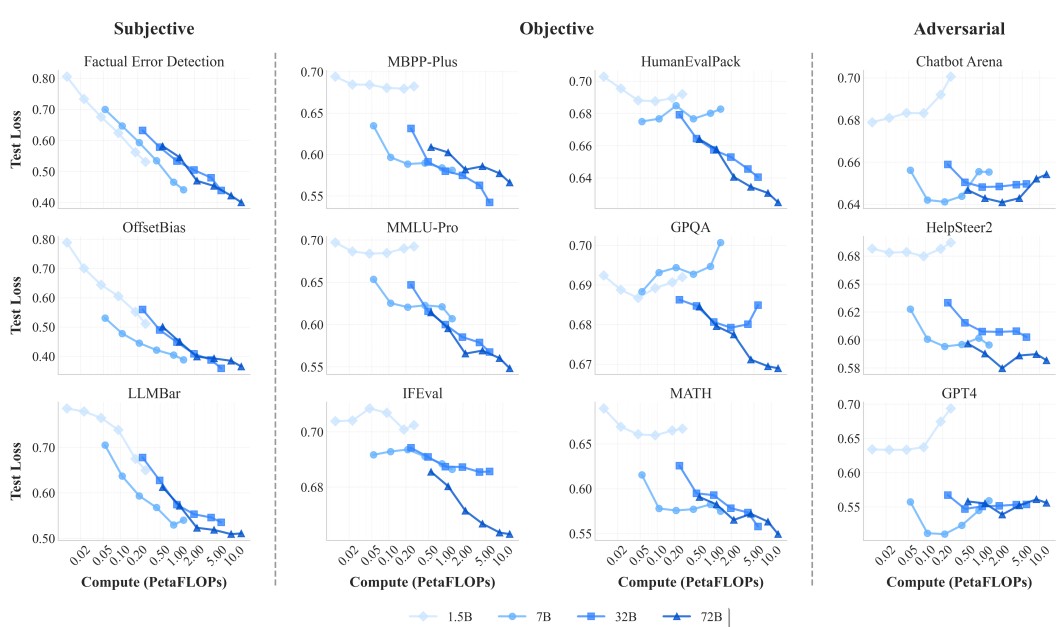

Figure 15: Compute-normalized scaling behavior across tasks. We compare performance trends under matched computational budgets. In general, increased computation leads to better performance, and certain tasks require larger base models in order to fully benefit from higher compute.

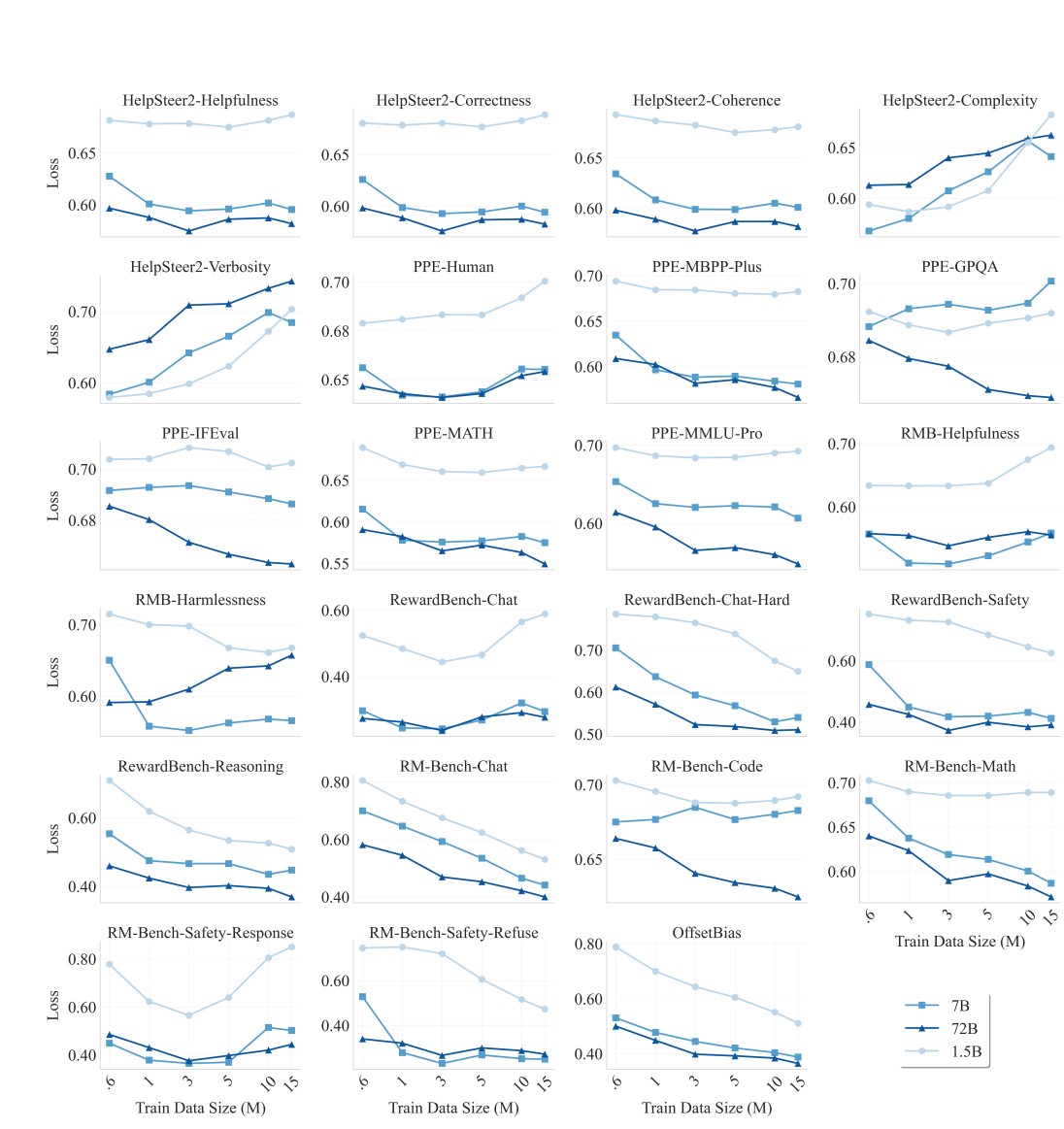

Figure 16: Loss trajectories across evaluation benchmarks as a function of training scale and model size, plotted on $log_2$ scale.

## C.4 Preference Fine-tuning Evaluation

Tables 7 and 8 present uncontrolled preference fine-tuning results for 7B and 72B models with and without WorldPM initialization, while Tables 9 and 10 show the style-controlled versions. The results align with findings in main text.

Notably, we include safety evaluations (RMB-Harmless, RewardBench-Safety, and RM-Bench-Safety), where the 72B model shows mixed performance without consistent improvement. This might be related to WorldPM's defensive behavior against pseudo-harmful content in safety domains. Detailed analysis can be found in Appendix C.3.

Table 7: Effect of WorldPM initialization on 7B model's PM fine-tuning performance across different evaluation categories (without style control). Abbreviations: RB - RewardBench, RM-B - RM-Bench, Safety-R/F - Safety-Response/Refuse.

| Metrics | HelpSteer2 | | UltraFeedback | | RLHFlow | |
|---|---|---|---|---|---|---|
| | w/o WorldPM | w/ WorldPM | w/o WorldPM | w/ WorldPM | w/o WorldPM | w/ WorldPM |
| **Subjective Evaluation** | | | | | | |
| PPE-Human | 62.05 | 63.80 (↑2.81%) | 60.88 | 61.64 (↑1.25%) | 65.54 | 66.03 (↑0.75%) |
| RMB-Helpful | 72.49 | 75.84 (↑4.62%) | 75.62 | 75.70 (↑0.11%) | 77.64 | 78.33 (↑0.88%) |
| RB-Chat | 97.49 | 97.21 (↓0.29%) | 97.49 | 97.21 (↓0.29%) | 98.04 | 97.77 (↓0.28%) |
| **Objective Evaluation** | | | | | | |
| PPE-MBPP | 67.07 | 71.31 (↑6.33%) | 66.60 | 72.24 (↑8.46%) | 71.97 | 74.44 (↑3.43%) |
| PPE-GPQA | 55.41 | 56.56 (↑2.08%) | 55.62 | 55.18 (↓0.81%) | 56.76 | 56.70 (↓0.10%) |
| PPE-IFEval | 57.50 | 56.17 (↓2.31%) | 56.21 | 56.64 (↑0.76%) | 61.87 | 62.54 (↑1.07%) |
| PPE-MATH | 70.80 | 71.74 (↑1.32%) | 71.64 | 73.77 (↑2.97%) | 78.79 | 79.94 (↑1.46%) |
| PPE-MMLU | 63.20 | 66.21 (↑4.76%) | 65.51 | 66.17 (↑1.01%) | 67.56 | 70.20 (↑3.90%) |
| RM-B-Code | 55.92 | 57.68 (↑3.14%) | 57.02 | 62.06 (↑8.85%) | 61.40 | 67.76 (↑10.36%) |
| RM-B-Math | 69.09 | 74.10 (↑7.25%) | 70.23 | 73.44 (↑4.58%) | 75.99 | 79.58 (↑4.73%) |
| RB-Reason | 81.62 | 96.12 (↑17.77%) | 83.19 | 91.86 (↑10.42%) | 94.13 | 96.16 (↑2.15%) |
| **Adversarial Evaluation** | | | | | | |
| RB-Chat-Hard | 62.50 | 74.89 (↑19.82%) | 61.51 | 67.32 (↑9.45%) | 66.01 | 69.96 (↑5.98%) |
| RM-B-Chat | 51.94 | 65.12 (↑25.37%) | 57.36 | 66.67 (↑16.22%) | 56.98 | 66.67 (↑17.01%) |
| OffsetBias | 63.76 | 82.40 (↑29.23%) | 72.98 | 78.17 (↑7.11%) | 73.54 | 80.50 (↑9.47%) |
| **Safety Evaluation** | | | | | | |
| RMB-Harmless | 68.41 | 73.58 (↑7.56%) | 69.22 | 70.23 (↑1.46%) | 66.78 | 66.84 (↑0.08%) |
| RB-Safety | 82.84 | 90.54 (↑9.30%) | 89.05 | 87.84 (↓1.37%) | 85.95 | 86.35 (↑0.47%) |
| RM-B-Safety-R | 87.90 | 92.99 (↑5.80%) | 95.54 | 94.27 (↓1.33%) | 91.08 | 93.63 (↑2.80%) |
| RM-B-Safety-F | 98.59 | 97.54 (↓1.07%) | 96.48 | 96.48 (0.00%) | 95.77 | 95.42 (↓0.37%) |

# D Details of Style Control

## D.1 Ablation Study on Style Control Factors

We control two style features: length and markdown formatting. As shown in Figure 17, we compare the impact on subjective evaluation performance when controlling length alone, markdown formatting alone, and both factors simultaneously. Our findings indicate that markdown formatting has less influence than length, further confirming that length is the primary style factor. Moreover, modeling both factors together enables more effective style control.

## D.2 Effect of Style Control Across Test Set Evaluation

Figure 19 and Figure 18 demonstrates the impact of style control on evaluation results (test accuracy and test loss) across all test sets. We observe several key findings:

- In HelpSteer2, the dimensions of Helpfulness, Correctness, and Coherence maintain substantial consistency. Complexity and Verbosity exhibit high correlation with length characteristics. Upon implementing length control, these two metrics show significant changes while maintaining stable trends.
- Some objective and robustness evaluation sets demonstrate notable shifts in performance (e.g., PPE-MBPP-Plus, PPE-GPQA, and RM-Bench-Chat). However, the performance gap between style-controlled and uncontrolled versions either remains constant or narrows

Table 8: Effect of WorldPM initialization on 72B model's PM fine-tuning performance across different evaluation categories (without style control). Abbreviations: RB - RewardBench, RM-B - RM-Bench, Safety-R/F - Safety-Response/Refuse.

| Metrics | HelpSteer2 | | UltraFeedback | | RLHFlow | |
|---|---|---|---|---|---|---|
| | w/o WorldPM | w/ WorldPM | w/o WorldPM | w/ WorldPM | w/o WorldPM | w/ WorldPM |
| **Subjective Evaluation** | | | | | | |
| PPE-Human | 62.22 | 65.53 (↑5.32%) | 62.09 | 64.51 (↑3.89%) | 67.63 | 68.29 (↑0.98%) |
| RMB-Helpful | 76.40 | 77.45 (↑1.38%) | 75.60 | 79.10 (↑4.64%) | 78.10 | 80.08 (↑2.53%) |
| RB-Chat | 97.91 | 98.88 (↑1.00%) | 97.21 | 96.09 (↓1.15%) | 98.32 | 98.04 (↓0.28%) |
| **Objective Evaluation** | | | | | | |
| PPE-MBPP | 70.19 | 77.45 (↑10.34%) | 74.32 | 78.65 (↑5.82%) | 81.00 | 81.85 (↑1.05%) |
| PPE-GPQA | 58.03 | 59.28 (↑2.15%) | 59.06 | 60.76 (↑2.88%) | 62.19 | 63.44 (↑2.01%) |
| PPE-IFEval | 58.96 | 63.50 (↑7.68%) | 60.16 | 63.83 (↑6.10%) | 63.32 | 65.04 (↑2.71%) |
| PPE-MATH | 69.73 | 76.64 (↑9.92%) | 72.58 | 77.89 (↑7.32%) | 79.10 | 82.52 (↑4.32%) |
| PPE-MMLU | 69.98 | 74.69 (↑6.73%) | 72.07 | 74.92 (↑3.96%) | 78.52 | 79.28 (↑0.97%) |
| RM-B-Code | 61.18 | 69.30 (↑13.26%) | 65.79 | 68.64 (↑4.33%) | 71.93 | 72.81 (↑1.22%) |
| RM-B-Math | 70.04 | 76.75 (↑9.58%) | 71.55 | 77.32 (↑8.06%) | 87.52 | 85.63 (↓2.16%) |
| RB-Reason | 96.61 | 98.22 (↑1.66%) | 95.70 | 96.44 (↑0.77%) | 96.33 | 97.38 (↑1.09%) |
| **Adversarial Evaluation** | | | | | | |
| RB-Chat-Hard | 69.41 | 80.26 (↑15.64%) | 75.66 | 77.63 (↑2.61%) | 74.12 | 75.33 (↑1.63%) |
| RM-B-Chat | 77.52 | 75.97 (↓2.00%) | 70.93 | 70.54 (↓0.55%) | 77.13 | 75.58 (↓2.01%) |
| OffsetBias | 76.29 | 85.79 (↑12.46%) | 84.47 | 85.86 (↑1.64%) | 83.18 | 81.90 (↓1.54%) |
| **Safety Evaluation** | | | | | | |
| RMB-Harmless | 68.96 | 70.10 (↑1.66%) | 68.58 | 67.93 (↓0.95%) | 63.70 | 62.65 (↓1.64%) |
| RB-Safety | 90.14 | 91.35 (↑1.35%) | 89.12 | 90.61 (↑1.67%) | 84.19 | 83.24 (↓1.12%) |
| RM-B-Safety-R | 94.59 | 94.27 (↓0.34%) | 97.45 | 98.09 (↑0.65%) | 94.90 | 93.63 (↓1.34%) |
| RM-B-Safety-F | 97.89 | 97.18 (↓0.72%) | 97.01 | 96.48 (↓0.54%) | 88.73 | 89.08 (↑0.40%) |

Table 9: Effect of WorldPM initialization on 7B model's PM fine-tuning performance across different evaluation categories. Abbreviations: RB - RewardBench, RM-B - RM-Bench, Safety-R/F - Safety-Response/Refuse.

| Metrics | Helpsteer2 | | UltraFeedback | | RLHFlow | |
|---|---|---|---|---|---|---|
| | w/o WorldPM | w/ WorldPM | w/o WorldPM | w/ WorldPM | w/o WorldPM | w/ WorldPM |
| **Subjective Evaluation** | | | | | | |
| PPE-Human | 63.32 | 64.93 (↑2.54%) | 63.22 | 63.86 (↑1.01%) | 66.18 | 66.78 (↑0.91%) |
| RMB-Helpful | 74.33 | 77.02 (↑3.61%) | 78.37 | 78.82 (↑0.58%) | 78.66 | 79.27 (↑0.78%) |
| RB-Chat | 97.21 | 97.07 (↓0.14%) | 97.49 | 97.21 (↓0.29%) | 98.04 | 97.77 (↓0.28%) |
| **Objective Evaluation** | | | | | | |
| PPE-MBPP | 72.24 | 73.55 (↑1.82%) | 69.77 | 73.55 (↑5.42%) | 75.06 | 76.83 (↑2.37%) |
| PPE-GPQA | 56.27 | 56.70 (↑0.76%) | 56.19 | 55.43 (↓1.36%) | 57.34 | 57.60 (↑0.44%) |
| PPE-IFEval | 58.81 | 57.34 (↓2.49%) | 57.42 | 57.03 (↓0.68%) | 62.09 | 62.38 (↑0.47%) |
| PPE-MATH | 71.04 | 71.64 (↑0.85%) | 71.76 | 74.36 (↑3.62%) | 78.98 | 80.16 (↑1.48%) |
| PPE-MMLU | 63.55 | 66.52 (↑4.67%) | 65.64 | 66.66 (↑1.55%) | 67.97 | 70.27 (↑3.39%) |
| RM-B-Code | 57.68 | 60.96 (↑5.70%) | 59.21 | 62.94 (↑6.30%) | 65.35 | 67.76 (↑3.69%) |
| RM-B-Math | 70.70 | 75.99 (↑7.49%) | 72.12 | 75.71 (↑4.98%) | 76.56 | 80.43 (↑5.06%) |
| RB-Reason | 88.02 | 96.44 (↑9.57%) | 92.35 | 95.39 (↑3.29%) | 95.91 | 96.82 (↑0.95%) |
| **Adversarial Evaluation** | | | | | | |
| RB-Chat-Hard | 81.25 | 83.99 (↑3.37%) | 77.52 | 79.82 (↑2.97%) | 81.47 | 82.57 (↑1.35%) |
| RM-B-Chat | 78.29 | 82.17 (↑4.95%) | 80.23 | 81.01 (↑0.97%) | 82.17 | 82.56 (↑0.47%) |
| OffsetBias | 78.87 | 86.60 (↑9.80%) | 81.68 | 82.94 (↑1.55%) | 83.43 | 85.69 (↑2.71%) |
| **Safety Evaluation** | | | | | | |
| RMB-Harmless | 72.73 | 73.99 (↑1.72%) | 69.74 | 70.92 (↑1.69%) | 62.86 | 63.79 (↑1.48%) |
| RB-Safety | 90.68 | 94.53 (↑4.25%) | 91.76 | 89.46 (↓2.50%) | 90.88 | 92.03 (↑1.26%) |
| RM-B-Safety-R | 90.45 | 93.95 (↑3.87%) | 96.50 | 96.82 (↑0.33%) | 92.99 | 96.18 (↑3.42%) |
| RM-B-Safety-F | 99.65 | 99.30 (↓0.35%) | 98.94 | 98.94 (0.00%) | 98.59 | 99.12 (↑0.54%) |
| **Alignment Evaluation** | | | | | | |
| Arena Hard | 65.88 | 70.27 (↑6.66%) | 67.93 | 72.12 (↑6.17%) | 72.41 | 74.62 (↑3.05%) |
| Alpaca Eval | 43.73 | 46.29 (↑5.84%) | 46.16 | 46.35 (↑0.40%) | 53.24 | 50.97 (↓4.27%) |
| Arena Hard-SC | 65.07 | 70.22 (↑7.91%) | 66.19 | 70.36 (↑6.30%) | 72.77 | 74.29 (↑2.09%) |
| Alpaca Eval-LC | 39.98 | 42.14 (↑5.41%) | 44.46 | 46.13 (↑3.76%) | 46.24 | 46.32 (↑0.17%) |

Table 10: Effect of WorldPM initialization on 72B model's PM fine-tuning performance across different evaluation categories. Abbreviations: RB - RewardBench, RM-B - RM-Bench, Safety-R/F - Safety-Response/Refuse.

| Metrics | Helpsteer2 | | UltraFeedback | | RLHFlow | |
|---|---|---|---|---|---|---|
| | w/o WorldPM | w/ WorldPM | w/o WorldPM | w/ WorldPM | w/o WorldPM | w/ WorldPM |
| **Subjective Evaluation** | | | | | | |
| PPE-Human | 64.76 | 66.68 (↑2.97%) | 65.12 | 66.94 (↑2.80%) | 68.14 | 68.93 (↑1.16%) |
| RMB-Helpful | 77.76 | 78.53 (↑1.00%) | 79.80 | 81.73 (↑2.41%) | 79.26 | 80.89 (↑2.05%) |
| RB-Chat | 98.04 | 98.88 (↑0.85%) | 97.63 | 96.37 (↓1.29%) | 98.46 | 97.91 (↓0.57%) |
| **Objective Evaluation** | | | | | | |
| PPE-MBPP | 70.50 | 78.76 (↑11.72%) | 75.02 | 79.46 (↑5.92%) | 82.28 | 82.39 (↑0.14%) |
| PPE-GPQA | 57.66 | 59.63 (↑3.42%) | 59.22 | 61.19 (↑3.33%) | 62.52 | 63.75 (↑1.97%) |
| PPE-IFEval | 59.24 | 63.63 (↑7.42%) | 60.57 | 64.34 (↑6.22%) | 64.26 | 65.39 (↑1.76%) |
| PPE-MATH | 69.94 | 77.17 (↑10.33%) | 72.77 | 78.16 (↑7.41%) | 79.36 | 83.01 (↑4.60%) |
| PPE-MMLU | 70.20 | 74.96 (↑6.79%) | 72.42 | 75.08 (↑3.67%) | 79.04 | 79.30 (↑0.32%) |
| RM-B-Code | 60.75 | 69.30 (↑14.08%) | 67.32 | 70.18 (↑4.23%) | 72.15 | 73.46 (↑1.82%) |
| RM-B-Math | 72.78 | 77.69 (↑6.75%) | 73.72 | 77.79 (↑5.51%) | 87.81 | 86.20 (↓1.83%) |
| RB-Reason | 96.65 | 98.25 (↑1.66%) | 96.30 | 97.48 (↑1.23%) | 97.52 | 98.15 (↑0.64%) |
| **Adversarial Evaluation** | | | | | | |
| RB-Chat-Hard | 84.87 | 87.28 (↑2.84%) | 84.54 | 83.00 (↓1.82%) | 84.21 | 84.54 (↑0.39%) |
| RM-B-Chat | 81.01 | 82.17 (↑1.44%) | 80.23 | 79.46 (↓0.97%) | 85.66 | 84.50 (↓1.36%) |
| OffsetBias | 83.14 | 89.76 (↑7.96%) | 87.29 | 88.05 (↑0.88%) | 87.89 | 87.47 (↓0.47%) |
| **Safety Evaluation** | | | | | | |
| RMB-Harmless | 69.42 | 70.34 (↑1.33%) | 69.57 | 68.43 (↓1.64%) | 60.47 | 59.96 (↓0.84%) |
| RB-Safety | 92.03 | 93.51 (↑1.62%) | 93.45 | 93.51 (↑0.07%) | 92.09 | 92.03 (↓0.07%) |
| RM-B-Safety-R | 95.54 | 97.13 (↑1.67%) | 98.41 | 98.41 (0.00%) | 98.41 | 97.13 (↓1.29%) |
| RM-B-Safety-F | 99.65 | 99.12 (↓0.53%) | 98.59 | 98.24 (↓0.36%) | 96.83 | 97.18 (↑0.36%) |
| **Alignment Evaluation** | | | | | | |
| Arena Hard | 88.52 | 89.72 (↑1.36%) | 89.16 | 90.37 (↑1.36%) | 90.41 | 90.82 (↑0.45%) |
| Alpaca Eval | 60.34 | 63.39 (↑5.07%) | 59.95 | 63.73 (↑6.31%) | 65.51 | 66.26 (↑1.14%) |
| Arena Hard-SC | 88.47 | 90.54 (↑2.34%) | 88.82 | 90.74 (↑2.16%) | 91.06 | 91.76 (↑0.77%) |
| Alpaca Eval-LC | 53.26 | 55.45 (↑4.12%) | 56.83 | 59.80 (↑5.23%) | 56.95 | 55.58 (↓2.40%) |

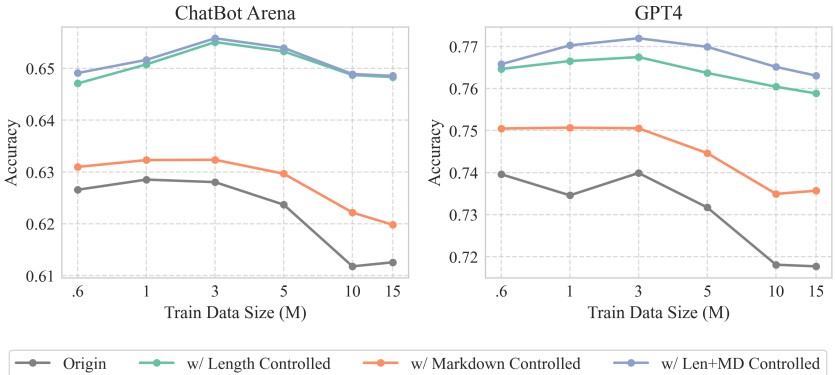

Figure 17: Impact of controlling different style factors on subjective evaluation performance.

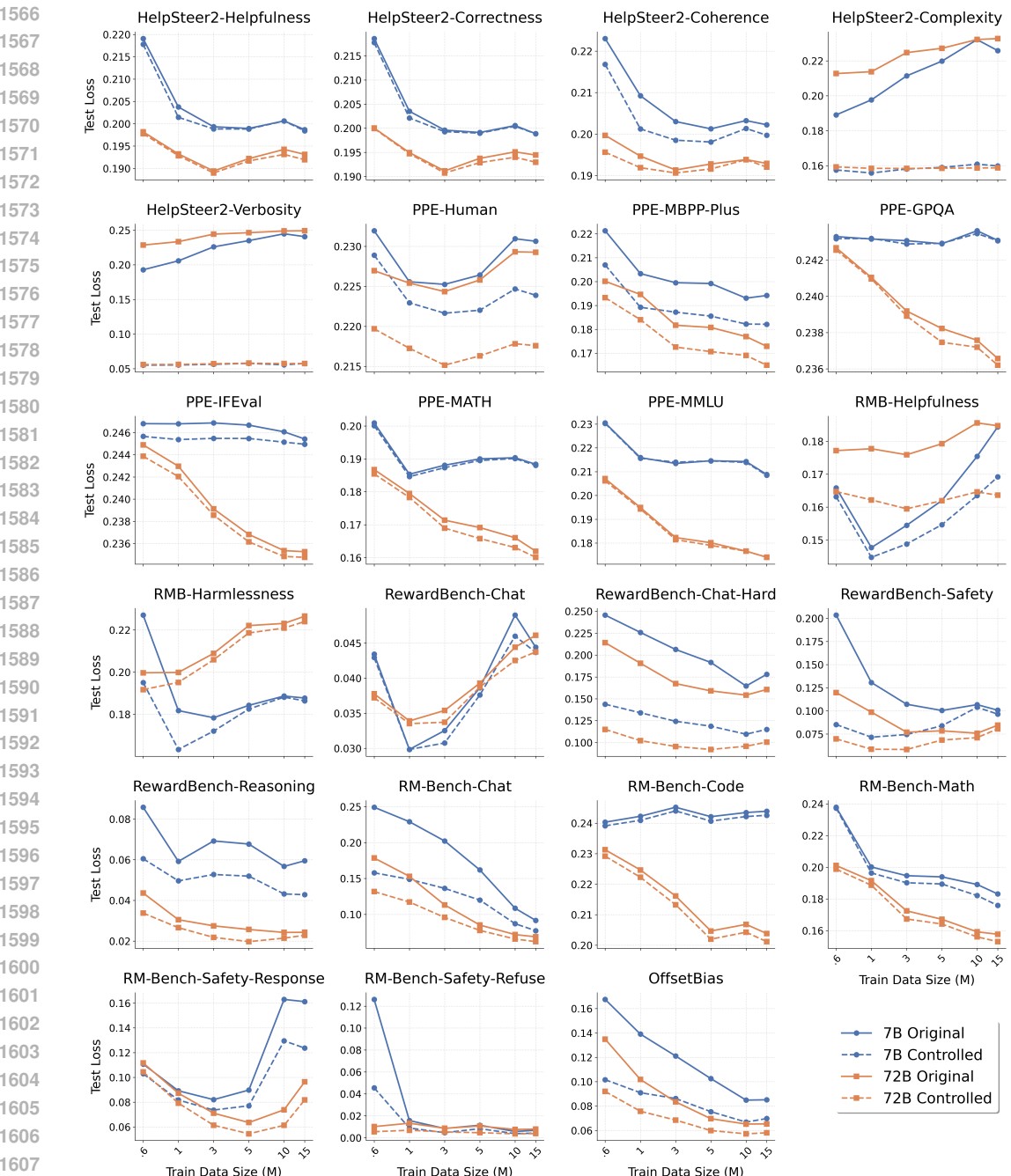

Figure 18: Effect of style control on test loss across all test set evaluation.

with increased training scale, contrasting with the widening gap in subjective domains. This suggests that modeling world preference gradually overcomes length bias, converging toward more accurate evaluation results.

- In the safety domain, RM-Bench-Safety-Refuse demonstrates exceptionally high accuracy even without style control, indicating WorldPM's inherent capability for safety discrimination. Other safety-related benchmarks show declining accuracy in later stages, potentially related to increased defense against pseudo-harmful queries.

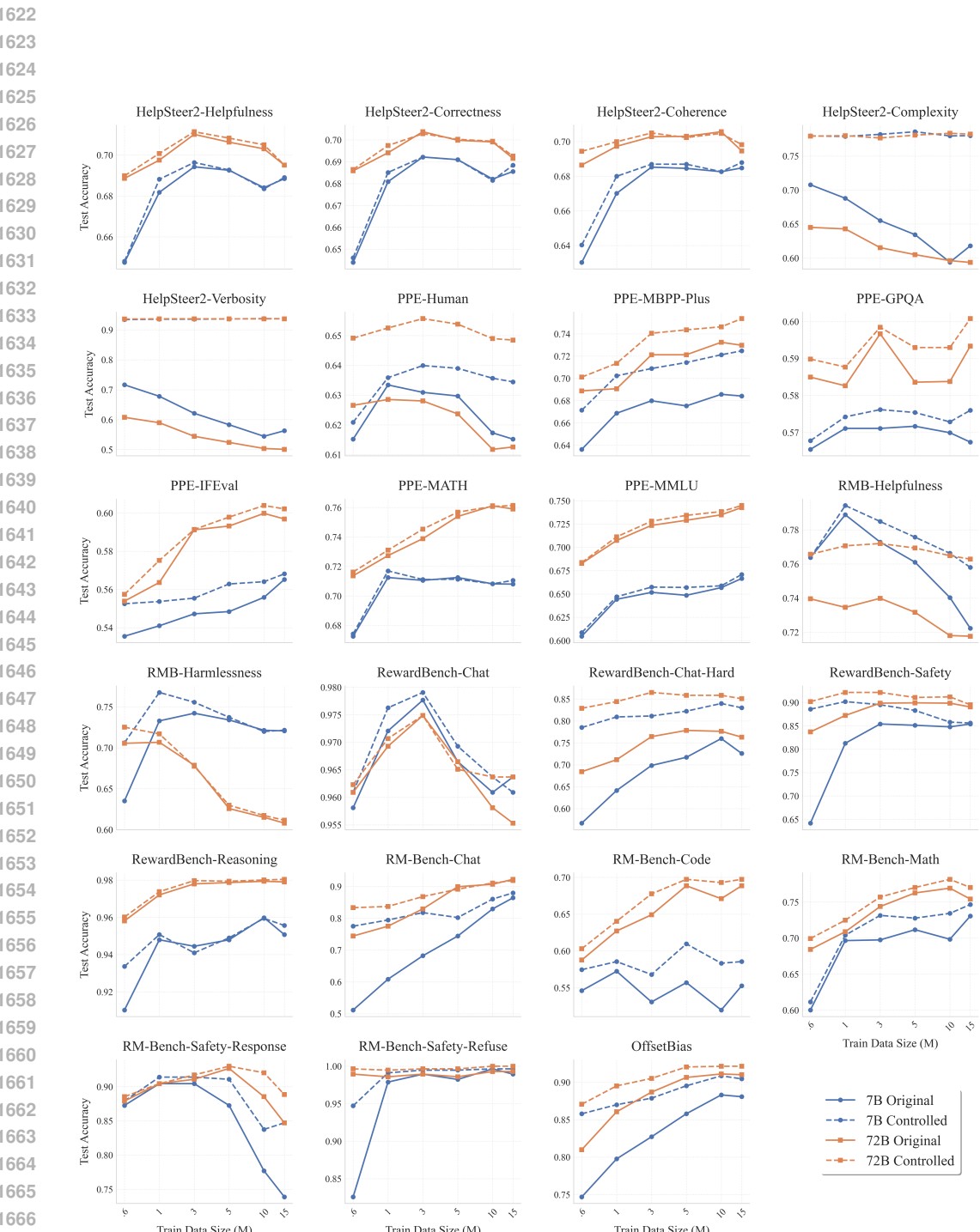

Figure 19: Effect of style control on performance across all test set evaluation.

## E  THE USE OF LARGE LANGUAGE MODELS (LLMs)

The use of LLMs in this paper is limited to polishing content and correcting errors during the writing phase. All content generated by LLMs has been thoroughly reviewed and verified.

## F  LIMITAITON

Our dataset comprises 15M preference pairs (approximately 30G tokens) from StackExchange, modest in contrast to conventional next-token prediction pre-training, which routinely utilizes datasets of trillion-token scale. Significant opportunities exist for expanding preference pretraining data through untapped sources such as various forums and social media platforms.

In addressing subjective evaluation biases, we control only for length and Markdown formatting. Many other influential factors, including emotional preferences and cultural tendencies, remain challenging to capture, underscoring the complexity of subjective assessment. The interconnected nature of various aspects in subjective evaluation makes comprehensive and granular assessment particularly challenging, as these elements prove difficult to isolate effectively.

## G  CASE STUDY

We present several examples from PPE-Human, including Figures 20, 21, and 22. In Figures 20 and 21, general humans chose incorrect answers, while WorldPM correctly identified the errors. Figures 22 demonstrates that data from general humans can be low-quality and uninformative, thus interfering with evaluation.

We also present several examples of training data used for modeling world preference, including Figures 23, 24, 25, and 26. These data often contain rich information, enabling the model to learn complex representations.

**Query**

Prove or disprove: for all integers $a$, $b$, $c$, $d$, if $a \mid b$ and $c \mid d$, then $(a + c) \mid (b + d)$.

**Chosen Response (World Preference Model Score: -1.84)**

This statement is true. This is because absolute values of sums of integers are equal to the sums of their absolute values.

In this case, since $a \mid b$ ($a$ divides $b$) and $c \mid d$ ($c$ divides $d$), then certainly $|b|$ and $|d|$ are both divisible by $|a|$ and $|c|$, respectively. So, setting $e = |a|$ and $f = |c|$, we have that both $b = q \cdot e$ for some integer $q$ and $d = r \cdot f$ for some integer $r$. Then, $b + d = (q + r) \cdot e + f$ is divisible by both $e$ and $f$, which are equal to $|a|$ and $|c|$. So $(b + d)$ is divisible by the sum $|a + c|$. Thus, $(a + c) \mid (b + d)$.

**Rejected Response (World Preference Model Score: 0.76)**

To prove or disprove the statement "for all integers $a$, $b$, $c$, $d$ if $a \mid b$ and $c \mid d$, then $a + c \mid b + d$", we can approach it by first understanding what the statement means and then testing it with specific examples.

**Definitions:** - $a \mid b$ means that there exists an integer $k$ such that $b = ak$. - $c \mid d$ means that there exists an integer $m$ such that $d = cm$.

**Claim:** - $a + c \mid b + d$ means there exists an integer $n$ such that $b + d = (a + c)n$.

We need to see if from $b = ak$ and $d = cm$, it follows that $b + d = (a + c)n$ for some integer $n$.

**Counterexample:** Let's consider specific numbers to see if this statement holds. - Choose $a = 2$, $b = 4$, $c = 3$, $d = 6$. - Clearly, $2 \mid 4$ because $4 = 2 \times 2$. - Clearly, $3 \mid 6$ because $6 = 3 \times 2$. - Now, $a + c = 2 + 3 = 5$ and $b + d = 4 + 6 = 10$.

We need to check if $5 \mid 10$. Since $10 = 5 \times 2$, in this case, $5$ does actually divide $10$. However, we need to check more examples to see if there are any that fail.

**Another Example:** - Choose $a = 2$, $b = 4$, $c = 3$, $d = 9$. - Again, $2 \mid 4$ since $4 = 2 \times 2$. - $3 \mid 9$ since $9 = 3 \times 3$. - Calculate $a + c = 2 + 3 = 5$ and $b + d = 4 + 9 = 13$.

Now, we need to check if $5 \mid 13$. Since $13$ is not divisible by $5$ (there is no integer $n$ such that $13 = 5n$), this proves that $5$ does not divide $13$.

**Conclusion:** The counterexample where $a = 2$, $b = 4$, $c = 3$, $d = 9$ disproves the statement. Even though $a$ divides $b$ and $c$ divides $d$, $a + c$ does not necessarily divide $b + d$. Thus, the statement "for all integers $a$, $b$, $c$, $d$ if $a \mid b$ and $c \mid d$, then $a + c \mid b + d$" is false.

Figure 20: Example from the PPE-Human test set where the chosen response's proof is wrong while the rejected response's proof is correct, and the world preference model identify this error.

**Query**

userscript to copy the text of the hovered link when the key combination alt+c is pressed. just present the code and make it concise.

**Chosen Response (World Preference Model Score: -0.66)**

```javascript
// @require     https://code.jquery.com/jquery-3.6.0.min.js

$(document).keydown(function(event){
    if(event.altKey && event.keyCode == 67){
        var link = $("hoveredElement").text();
        clipboard.writeText(link);
    }
});
```

**Rejected Response (World Preference Model Score: 1.42)**

```javascript
// ==UserScript==
// @name        Copy Hovered Link Text
// @namespace   http://tampermonkey.net/
// @version     1.0
// @description Copy the text of the hovered link when Alt+C is
     pressed
// @author      You
// @match       *://*/*
// @grant       none
// ==/UserScript==

(function() {
    'use strict';
    let hoveredLinkText = '';
    document.addEventListener('mouseover', function(e) {
        if (e.target.tagName.toLowerCase() === 'a') {
            hoveredLinkText = e.target.innerText || e.target.
                textContent;
        }
    });
    document.addEventListener('keydown', function(e) {
        if (e.altKey && e.key === 'c') {
            if (hoveredLinkText) {
                navigator.clipboard.writeText(hoveredLinkText).then
                    (() => {
                    console.log('Text copied to clipboard: ',
                        hoveredLinkText);
                }).catch(err => {
                    console.error('Failed to copy text: ', err);
                });
            }
        }
    });
})();
```

Figure 21: Example from the PPE-Human test set, where Chosen uses an invalid selector $("hoveredElement"), while Rejected provides a correct solution.

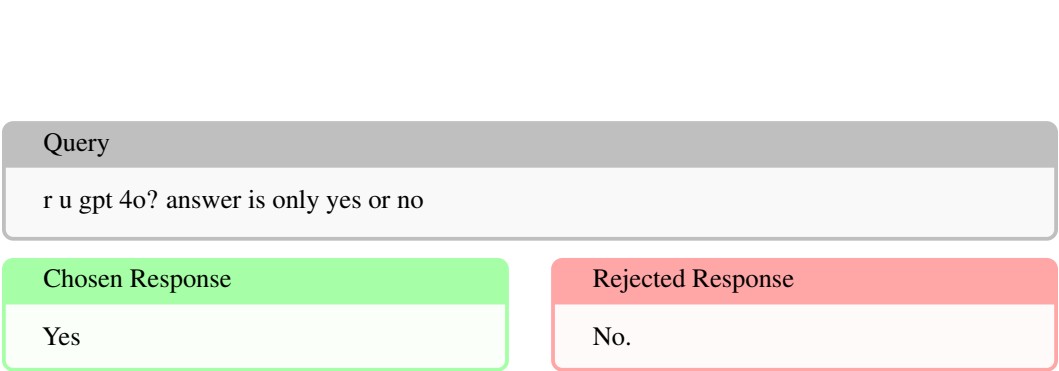

Figure 22: Example from the PPE-Human test set, where the relative quality cannot be determined.

**Query (From freelancing.stackexchange)**

Title: Where can a front end web developer start freelancing?
Let's say you're beginning web development, and you want to start earning money. Where do you get started?
I know a portfolio would be a start, but I don't know where to apply just as a front-end developer for jobs.
I don't know and don't want to know PHP, Ruby, Rails, and other stuff. I just want to do front end development part time, because I find it interesting.
By the way, if it helps, I live in NYC.

**Chosen Response (Upvotes: 11)**

The challenge that you're going to run into with Freelance web development is that, in my experience, many Web developers have pretty broad understanding of the entire development stack from databases to server-side coding to front-end HTML, CSS, and JavaScript.
Thus, if you're just starting out and wish to avoid learning other parts of the stack, you may run into trouble finding work. In my experience, when a Freelancer needs to outsource something, it's because he or she needs a specialist in that area.
Using myself as an example, I lack knowledge in design, so I'm less likely to find work building websites since most people who build websites can also manage to do the little bit of scripting necessary to make the site functional. However, I specialize in JavaScript. If you have a Web application and are stuck somewhere or are doing something complicated, that's where I come in.
If you're going to specialize in nothing but the frontend, you're going to need to specialize in it, which means you're going to need to be rally good, write really clean, W3C-valid HTML code that works in every browser, and have a deep understanding of front-end User Experience.
Even then, you need to understand at least a little server-side code. The best front-end coders I've ever worked with still knew a little PHP so that we could understand how we were going to integrate things. They weren't experts in it, but they weren't ignorant to it either; it's naive to think that you can get away with never ever knowing how the server-side works.
With that said, you have to start somewhere, and the best way for you to do that is to look for small jobs that require just minor changes to existing front-end code. This will help build your skill-set in that area while hopefully giving you some income.

**Rejected Response (Upvotes: 8)**

If you're doing front end development, you will need a portfolio before you get any big jobs, end of story. So how do you get a portfolio without the jobs?
What I did before was make sample pages for friends of the family, and a couple volunteer organisations I belonged to. It was all under the impression that they only paid if they used my designs, but accept that it's just a start. I could at least get some styles that I could do.
That portfolio should be online though on your personal website, so potential customers can see your skill, design, and technique.
Again, just make designs, no matter who for, and make sure they show your best talents

Figure 23: Example 1 drawn from StackExchange training dataset.

**Query (From physics.stackexchange)**

Title: Could a non-rotating planet, in a binary situation, have gravity?
Is it possible for a singular planet to rotate around a single sun in the same way that our moon rotates around the Earth? If so, would it be possible for that planet, if as large or larger than Earth, to have gravity?

**Chosen Response (Upvotes: 4)**

Yes. Literally everything that has mass (and even some things that don't) will "have gravity". It is certainly possible for a single planet to orbit a single star (as opposed to our planet, for example, which is accompanied by the other planets of the solar system). In fact, if anything, a single planet will be *more* stable because it will not be perturbed by other massive bodies. But in all of these cases, gravity will certainly be present. For example, the reason a planet will even stick together to *be* a planet is because gravity holds it together.

**Rejected Response (Upvotes: 0)**

Responding directly: Yes it is possible and is expected in cosmological time scale.
It is expected for both bodies to have the same face to the other with sufficient time, but the one of smaller mass will generally slow its rotation with respect to the body of major mass first, as strictly that phenomenon is connected to the angular moment of rotation, that moment is proportional to mass and speed of rotation, but other factors influence the loss of rotation (braking), such as the existence of fluids in the celestial body (plasma in the stars and atmosphere and water or other fluid in the planets and satellites, the denser the braking of the celestial body, the energy in this case becomes tidal energy in the fluids. What speeds up the process is proximity. The body that will first present the same face is the one that has the lowest energy (moment) and the set of conditions that predispose them to lose energy faster, such as less mass, lack of atmosphere and oceans.
The earth does not yet have the same face for the Moon, but millions of years ago the day on earth lasted only about 10 hours, from here a few million years ago when the day on earth lasts for a lunar month, both the Moon and Earth will show the same face and will be synchronized. In his question the case is of the existence of a single planet in the system, this only facilitates the synchronization because there is less gravitational perturbation of other celestial bodies of great mass, but remembering that the proximity is the main factor in the speed of the process.
See Wikipedia article: Gliese 581c
Search the internet for these titles for more information: Tidal coupling (Tidal Locking)
You will see that the Pluto system and its moon Charon are already synchronized each showing the same face to the other component of the system and there is a planet in the star Tau Bootes that presents this synchronization.
There is also an excellent article on the subject in the book "Asimov explains" by Isaac Asimov
Answering the second question:
Yes, the planet independent of size will have gravity, but the resulting value will be different on each side. On the side permanently presented to the star will be smaller than on the other side but will not be zero. There will be a point between the star and the planet where gravity will vanish, but it will still be off the planet. The loss of momentum of rotation partly alters the moment of translation, perhaps making it happen to the planet what already happens to our Moon that moves away from Earth at a speed of 3.78 centimeters per year, with the possibility of tearing from the earth.
Excuse my english please.

Figure 24: Example 2 drawn from StackExchange training dataset.

**Query (From tex.stackexchange)**

Title: How to add custom C++ keywords to be recognized by Minted?
I have defined several structures and types in a C++ project. I have to write some documentation for this code, and I use the minted package to typeset my code.
I would like to be able to add some keywords to the minted database, so minted will also recognize these and give them the appropriate color. (I would like to have them colored in the same way a type, eg. double would be colored.)
[1]: http://www.ctan.org/pkg/minted

**Chosen Response (Upvotes: 14)**

Minted uses Pygments to parse source-code, so one way to go about this is by modifying pygments/lexers/compiled.py and adding your types there (Somewhere about line 210 looks reasonable to me). This is what I had to do when Pygments was incorrectly parsing a symbol in some Erlang I'd written.
The obvious disadvantage of this is that Pygments is now non-standard. This will mean a number of things:
- This change has to be made on all PCs that want to compile this document, meaning the document will be built inconsistantly or lots of people have to have modified Pygments installations. - Future updates to Pygments could break your modifications and so your document may not be correctly syntax-highlighted in the future. - You may incorrectly parse/highlight future documents if you use these keywords in a different way (So it may print a variable name in the same way it prints a type, for example).

**Rejected Response (Upvotes: 4)**

Another (obvious) solution would be to write your own lexer (i.e. a superset of c++)
[1]: http://pygments.org/docs/lexerdevelopment/

Figure 25: Example 3 drawn from StackExchange training dataset.

**Query (From math.stackexchange)**

Title: Does every continuous map from $\mathbb{Q}$ to $\mathbb{Q}$ extends continuously as a map from $\mathbb{R}$ to $\mathbb{R}$?
Given a continuous function $f : \mathbb{Q} \to \mathbb{Q}$ ,does there exist a continuous function $g : \mathbb{R} \to \mathbb{R}$ such that $g|_{\mathbb{Q}} = f$?
What I have no Idea about how to attempt this Question! Any suggestion will be very helpful.

**Chosen Response (Upvotes: 16)**

Not in general. Consider the function $f_0 : \mathbb{Q} \to \mathbb{Q}$ defined by

$$f_0(x) = \begin{cases} 0 & \text{if } x^2 < 2 \\ 1 & \text{if } x^2 > 2 \end{cases} \quad .$$

It is easy to see that $f_0$ is continuous on $\mathbb{Q}$ (since $\sqrt{2} \notin \mathbb{Q}$), but if $g_0 : \mathbb{R} \to \mathbb{R}$ is an extension of $f_0$ to $\mathbb{R}$, then $g_0$ is necessarily discontinuous at $\sqrt{2}$.
It is worth noting that if $f : \mathbb{Q} \to \mathbb{Q}$ can be extended to a continuous function $g : \mathbb{R} \to \mathbb{R}$, then such an extension is unique. That is, if $g : \mathbb{R} \to \mathbb{R}$ and $h : \mathbb{R} \to \mathbb{R}$ are continuous functions such that $g(x) = h(x)$ for all $x \in \mathbb{Q}$, then $g = h$. This is because $\mathbb{Q}$ is a dense subset of $\mathbb{R}$.

**Rejected Response (Upvotes: 6)**

Yes, it's possible, iff $f$ is sequentially continuous, in the sense that if $a_n$ and $b_n$ are two sequences of rational numbers that converge to the same real number, then $f(a_n)$ and $f(b_n)$ are two converging sequences that converge to the same real number.
If you want a characterisation that doesn't mention $\mathbb{R}$ at all, you probably have to use the notion of Cauchy: "$a_n$ and $b_n$ are Cauchy sequences such that $a_n - b_n$ converges to 0".

Figure 26: Example 4 drawn from StackExchange training dataset.

