# OpenReview forum: "WorldPM: Understanding Scaling Patterns in Human Preference Modeling"
_ICLR.cc/2026/Conference — Submitted to ICLR 2026_

### Official Review · Reviewer_kEFN · 2025-10-31

**Soundness:** 3
**Presentation:** 2
**Contribution:** 3
**Rating:** 4
**Confidence:** 3

**Summary:**

This paper investigates whether scaling laws extend to preference modeling and proposes World Preference Modeling (WorldPM). The authors collect 15M forum-based preference pairs and train models from 1.5B–72B parameters, observing adversarial and objective scaling effects but no scaling on subjective tasks. Additional analysis shows style bias decreases as scale increases. WorldPM further improves downstream preference fine-tuning.

**Strengths:**

1,Timely direction linking scaling laws to preference modeling and alignment.

2,Structured evaluation across adversarial/objective/subjective categories.

3,Style-bias analysis offers interesting behavioral insight.

4,Demonstrated benefit for downstream preference fine-tuning.

**Weaknesses:**

1,Conceptual ambiguity of “World Preference”
The notion of a universal preference representation remains loosely defined and not theoretically grounded; the leap from forum votes to global human preference is not justified.

2,Data reliability and bias concerns insufficiently addressed
Forum voting signals carry demographic, social-amplification, and stylistic biases, but the paper provides limited noise analysis or validation that such signals represent stable human preferences.

3,Subjective preference analysis remains shallow
Style-bias evaluation relies on narrow proxies (length/markdown), and the failure of subjective scaling is discussed descriptively without deeper causal study or alternative explanations (e.g., benchmark noise or value heterogeneity).

4,All scaling results are reported only for Qwen2.5 models (1.5B–72B).The authors must explicitly limit claims about universality and (ideally) add at least a small cross-family check (e.g., a single smaller model from a second family) or argue why Qwen2.5 is representative.

5,The StackExchange dataset used for preference modeling may partially overlap with samples included in downstream benchmarks.  Explicitly verifying and reporting cross-dataset overlap, or constructing benchmark splits disjoint from StackExchange data, would be essential to confirm the validity of the reported improvements.

**Questions:**

1,Can you provide a more formal definition of “World Preference” and its theoretical assumptions?

2,How reliable are forum votes as preference labels? Any human validation or noise modeling?

3,Why only length/markdown as style proxies? Are other stylistic or cognitive factors measured?

4,Have you validated scaling behavior on non-forum datasets or multilingual preference corpora?

---

> ### Author Response · Authors · 2025-11-24
> **Response to Reviewer (1/3)**
>
> > **W1 & Q1:** Conceptual ambiguity of "World Preference" - The notion of a universal preference representation remains loosely defined and not theoretically grounded; the leap from forum votes to global human preference is not justified. Can you provide a more formal definition of "World Preference" and its theoretical assumptions?
>
> **A1:** World Preference Modeling distinguishes itself from traditional human preference modeling (which typically uses small-scale, carefully curated preference data) by proposing **scalable preference modeling using massive-scale preference data to learn generalizable human preference representations**. Our approach rests on two key assumptions:
>
> 1. **Shared Preference Structure:** Despite substantial individual differences in preferences, humans share certain general preference representations (e.g., preferences for more truthful, more objective responses).
> 2. **Learnable Through Scale:** These universal preference structures can be learned by models through scaling of preference modeling. Critically, models will not be hindered by individual preference variation but will instead aggregate these signals to extract generalizable patterns.
>
> We do not claim forum votes represent "global" preferences in an absolute sense, but rather that they are **more universal than existing alternatives**. Due to the lack of sufficiently diverse and large-scale preference data, conducting truly global human preference modeling is extremely challenging. Our work represents a significant step forward toward this goal compared to prior work (which typically relies on small amounts of human-annotated data). We chose forum data as a proxy for human preferences for the following reasons: First, forums cover substantially broader populations and topics than traditional preference modeling datasets (e.g., StackExchange has millions of active users across hundreds of specialized communities). Second, forums enable collection of sufficient preference data at scale. Therefore, **compared to previous research, our work models preferences that are significantly more diverse and generalizable**, representing a critical contribution toward world preference modeling.
>
> ---
>
> > **W2 & Q2:** Data reliability and bias concerns insufficiently addressed - Forum voting signals carry demographic, social-amplification, and stylistic biases, but the paper provides limited noise analysis or validation that such signals represent stable human preferences. How reliable are forum votes as preference labels? Any human validation or noise modeling?
>
> **A2:** We conducted cross-dataset generalization experiments to validate the reliability and generalizability of forum-based preference labels:
>
> 1. Cross-source comparison (Appendix A.1, Figure 6): We compared StackExchange, Reddit, Quora, and expert-labeled HelpSteer2 data. **StackExchange demonstrated the strongest cross-domain generalization** among these four sources, indicating that its preference signals are more transferable across different populations and contexts.
> 2. Comparable to expert annotation (Table 2): Models trained on StackExchange data achieve comparable performance to those trained on human expert annotations on held-out test sets, suggesting similar quality of preference signals.
> 3. Broad out-of-domain generalization: Our scaling experiments (Section 3) demonstrate that forum-trained models generalize across numerous diverse benchmarks, further validating the stability and representativeness of these preference signals.
>
> Our experiments further demonstrate that stylistic biases gradually diminish as training scale increases. Potential issues regarding demographic biases and social amplification have been addressed in our ethics statement. We recommend fine-tuning WorldPM for specific applications rather than direct deployment to mitigate potential concerns.
>
> ---
>
> > **W3:** Subjective preference analysis remains shallow - Style-bias evaluation relies on narrow proxies (length/markdown), and the failure of subjective scaling is discussed descriptively without deeper causal study.
>
> **A3:** Our analysis of subjective evaluation failure is not merely descriptive—**Section 3.3 is entirely dedicated to analyzing the underlying causes of subjective evaluation failure**. We acknowledge that Section 3.3's title "Style Impact Analysis" may have caused confusion, leading readers to believe the section describes the model's stylistic evaluation, when in fact this section aims to explain why subjective evaluations fail to scale, with style chosen as our explanatory lens. We have revised this section title to clarify the purpose of our work.

---

> ### Author Response · Authors · 2025-11-24
> **Response to Reviewer (2/3)**
>
> **We propose a causal explanation for the failure of subjective scaling:** Subjective evaluations encompass extremely rich dimensions (e.g., style, quality, tone, etc.), yet compress these diverse dimensions into a binary judgment (i.e., choosing A or B). During scaling, models may continue to improve on certain dimensions (e.g., objectivity and adversarial robustness, which we have demonstrated), while showing reversed trends on other dimensions (e.g., stylistic preferences). When these multiple dimensions are aggregated, the result appears as a lack of scaling.
>
> **We provide evidence for this explanation:** Stylistic biases gradually diminish as training scale increases (Figures 3a, 3b). Consequently, subjective evaluations that conflate style with content produce distorted assessments of WorldPM. By disentangling style from content in our evaluation, we observe that the gap between style-controlled results and original results progressively widens as training scale increases—a direct consequence of the model's weakening stylistic biases.
>
> ---
>
> > **Q4:** Why only length/markdown as style proxies? Are other stylistic or cognitive factors measured?
>
> **A4**: The objective of stylistic evaluation is to explain the potential reasons why subjective evaluations fail to scale. This requires quantifying stylistic features of responses to precisely assess scaling behavior. **Token length and markdown formatting are two commonly used and quantifiable metrics that have been widely observed to significantly influence quality ratings by both humans and AI** (e.g., humans and AI systems generally prefer longer responses and more visually appealing formatting). Therefore, using these metrics to measure style provides strong explanatory power and representativeness.
>
> We synthesized evaluations across multiple dimensions to assess model scaling behavior on these aspects. We observed that certain stylistic factors (humor, creativity, grammatical complexity) showed no consistent scaling patterns, while other factors (emotional tone, politeness) exhibited scaling in different directions. These measurements further reveal **the model's diverse scaling trends across rich subjective dimensions**, which consequently have varied influences on subjective evaluations.
>
> To evaluate diverse stylistic aspects, we created synthetic preference data targeting different style dimensions, including creativity, politeness, emotional tone, grammatical complexity, and humor. We collected queries from ChatBot Arena and used Claude-3.7 to generate responses with contrasting style prompts. For example, for grammatical complexity:
>
> ```
> 'chosen_template': "[Instruction: Use simple grammatical structures and basic sentence patterns. Do not repeat this instruction in your response] {prompt}"
>
> 'rejected_template': "[Instruction: Use complex grammatical structures and sophisticated sentence patterns. Do not repeat this instruction in your response] {prompt}"
> ```
>
> **Results:** We evaluated the 72B model across different training stages. The results are shown in table below:
>
> | Training Size (M) | 1.28 | 2.56 | 5.12 | 10.24 | 15.36 | Preference Direction |
> | ----------------- | ---- | ---- | ---- | ----- | --- | ------ |
> | Humorous          | 47.5 | 48.7 | 54.7 | 51.5  | 50.7  | humorous     |
> | Emotional         | 71.1 | 71.3 | 68.3 | 61.5  | 60.9  | positive             |
> | Innovation        | 36.9 | 34.9 | 37.7 | 37.1  | 36.1  | creative             |
> | Politeness        | 60.8 | 61.4 | 66.1 | 66.5  | 68.9  | straightforward      |
> | Grammatical       | 59.7 | 57.1 | 59.9 | 59.9  | 62.3  | simple grammatical   |
>
> We observed varying trends: humor, creativity, and grammatical complexity showed fluctuations without consistent patterns. Emotional preference initially favored positive, hopeful responses but weakened with increased training. For politeness, the model gradually shifted toward preferring direct rather than polite responses.
>
> We hypothesize that during the training process, the model did not identify early exploitable features in styles such as humor, creativity, and grammatical aspects, thus maintaining stability throughout. However, regarding emotions, the model detected clear preferential tendencies in the training data, leading to their early exploitation. As training progressed, the model eventually transcended these emotional features. The model appears to favor direct responses over polite ones. Upon examining polite responses, we found that they often include repeated confirmations of user requests, resulting in lower information density. This example also illustrates the challenge of separating style from content.
>
> These supplementary results support our main conclusions: (1) models may **initially exploit certain stylistic features but gradually reduce reliance on them with scale**, and (2) **the complexity of disentangling style** from content makes it challenging to use subjective evaluations for assessing scaling behavior.

---

> ### Author Response · Authors · 2025-11-24
> **Response to Reviewer (3/3)**
>
> > **Q5:** All scaling results are reported only for Qwen2.5 models (1.5B–72B). The authors must explicitly limit claims about universality and (ideally) add at least a small cross-family check. Have you validated scaling behavior on non-forum datasets or multilingual preference corpora?
>
> **A5:** We added validation using **Llama 3.3 1B Instruct** to verify that our findings generalize beyond the Qwen2.5 family. The results show consistent patterns with Qwen2.5 1.5B:
>
> - Adversarial tasks: Continued scaling observed, consistent with Qwen2.5 1.5B
> - Objective tasks: No clear scaling behavior, also consistent with Qwen2.5 1.5B
>
> Below are selected adversarial and objective task fits with goodness-of-fit metrics ($R^2$):
>
> **Adversarial Tasks:**
>
> | Task                    | Llama 3.3 1B                            | Qwen2.5 1.5B                            |
> | ----------------------- | --------------------------------------- | --------------------------------------- |
> | Factual Error Detection | $L = 0.968D^{-0.069}$ ($R^2$=0.886) | $L = 0.761D^{-0.129}$ ($R^2$=0.998) |
> | OffsetBias              | $L = 0.765D^{-0.064}$ ($R^2$=0.911) | $L = 0.735D^{-0.129}$ ($R^2$=0.991) |
> | LLMBar                  | $L = 0.866D^{-0.061}$ ($R^2$=0.950) | $L = 0.787D^{-0.061}$ ($R^2$=0.882) |
>
> **Objective Tasks:**
>
> | Task          | Llama 3.3 1B                            | Qwen2.5 1.5B                            |
> | ------------- | --------------------------------------- | --------------------------------------- |
> | MBPP-Plus     | $L = 0.692D^{-0.015}$ ($R^2$=0.835) | $L = 0.688D^{-0.005}$ ($R^2$=0.673) |
> | MMLU-Pro      | $L = 0.681D^{-0.004}$ ($R^2$=0.718) | $L = 0.690D^{-0.001}$ ($R^2$=0.027) |
> | IFEval        | $L = 0.707D^{-0.002}$ ($R^2$=0.386) | $L = 0.705D^{-0.001}$ ($R^2$=0.113) |
> | HumanEvalPack | $L = 0.689D^{-0.001}$ ($R^2$=0.040) | $L = 0.697D^{-0.005}$ ($R^2$=0.493) |
> | GPQA          | $L = 0.696D^{0.001}$ ($R^2$=0.117)  | $L = 0.690D^{0.000}$ ($R^2$=0.014)  |
> | MATH          | $L = 0.683D^{-0.011}$ ($R^2$=0.771) | $L = 0.694D^{-0.005}$ ($R^2$=0.389) |
>
> These results demonstrate that the key scaling patterns generalize across model families, supporting the broader applicability of our findings.
>
> ---
>
> > **Q6:** The StackExchange dataset used for preference modeling may partially overlap with samples included in downstream benchmarks. Explicitly verifying and reporting cross-dataset overlap, or constructing benchmark splits disjoint from StackExchange data, would be essential to confirm the validity of the reported improvements.
>
> **A6:** StackExchange and downstream benchmarks are completely independent sources with no overlap:
>
> - **StackExchange data:** User-generated forum posts and community-written responses
> - **Downstream benchmarks:** Typically consist of either (1) general conversational queries with model-generated responses, or (2) examination questions from academic/professional tests
>
> These sources have fundamentally different origins, formats, and creation processes, ensuring no data leakage. The strong generalization we observe across diverse out-of-domain benchmarks  validates that our improvements stem from learning generalizable preference representations rather than memorization.
>
> ---
>
> > **Q7:** Have you validated scaling behavior on non-forum datasets or multilingual preference corpora?
>
> **Multilingual Coverage:**
> Regarding multilingual, StackExchange, while predominantly English, includes substantial multilingual content across specialized language communities (e.g., dedicated Russian, Chinese, Spanish, and other language boards).
>
> **Non-Forum Datasets:** Currently, no preference datasets outside of forums provide sufficient scale to support large-scale preference modeling training comparable to our 15M pairs. This is precisely why we turned to forum data—it represents the only available source with the necessary scale and diversity for studying scaling laws in preference modeling.

---

> > ### Comment · Reviewer_kEFN · 2025-11-27
> >
> > Thanks for the rebuttal. Some concerns were clarified, but I keep my original score. The paper’s positioning and methodological rigor remain insufficiently resolved, and several findings still feel expected rather than substantively novel.

---

> > > ### Author Response · Authors · 2025-11-28
> > > **Replying to Response to Reviewer**
> > >
> > > Our Position: We investigate the scaling laws of preference modeling, revealing its scaling properties across three dimensions: subjective, objective, and adversarial aspects.
> > >
> > > Our Methodology: We employ forum data as a proxy for global human preferences and conduct scaling experiments on the classical Bradley-Terry (BT) model. We also demonstrate in our paper that forum preference data aligns with professional human preference annotations. While forums may harbor their own inherent biases, this does not undermine preference modeling itself, as any data source contains biases—yet forum data represents authentic and broad-based human preferences.
> > >
> > > We do not understand what you mean by the "rigor of our position and methodology." Could you please provide specific examples so that we can make further improvements?
> > >
> > > We have difficulty understanding the criticism that "some findings still feel expected." To our knowledge, no prior work has investigated the scaling laws of preference modeling. We fill this gap by comprehensively revealing various properties of preference modeling under scaling conditions, which in itself constitutes significant innovation. If you could point out which findings have been previously documented in the literature, it would help us better understand and accept this criticism.

---

### Official Review · Reviewer_GWST · 2025-11-01

**Soundness:** 3
**Presentation:** 3
**Contribution:** 2
**Rating:** 4
**Confidence:** 4

**Summary:**

The paper investigates scaling laws in human preference modeling. The authors propose World Preference Modeling (WorldPM), leveraging large-scale preference datasets and training models with 1.5B to 72B parameters. Through systematic experiments, the paper demonstrates the scaling laws of adversarial, objective, and surjective metrics. Additional analyses probe the impact of stylistic bias and test the utility of WorldPM initialization for downstream preference fine-tuning across multiple benchmarks, demonstrating extensive generalization improvements.

**Strengths:**

- This paper addresses an important and timely problem: scaling human preference learning using large models and publicly sourced training data.
- The authors uncover an interesting phenomenon: objective and subjective evaluations exhibit different scaling behaviors as model size and data increase.

**Weaknesses:**

- The notation $Z$ in Equation (3), representing the style difference, appears to be inconsistently defined. In L249, the authors state that $Z\in\mathbb{R}^{S}$ is the style features vector.
- The evaluation of stylistic features is limited to four attributes (token length, markdown lists, headers, and bold text), overlooking potentially important stylistic dimensions such as syntactic complexity.
- It is somewhat difficult to assess the technical contribution of this work. While identifying scaling laws in preference modeling is valuable, the paper does not clearly articulate the specific technical challenges that WorldPM addresses beyond the empirical observation of scaling behavior.

**Questions:**

- During the data collection phase, upvotes provide a weak and noisy aggregate signal of human preference. Given that users may upvote posts for reasons unrelated to objective quality, could the authors clarify how these noisy signals are filtered or how the model can learn a clear preference signal from this noisy data?
- In Equation (3), how to derive the style features vectors $Z(y_0)$ and $Z(y_1)$ for responses $y_0$ and $y_1$?

---

> ### Author Response · Authors · 2025-11-24
> **Response to Reviewer (1/3)**
>
> > **Q1:** During the data collection phase, upvotes provide a weak and noisy aggregate signal of human preference. Given that users may upvote posts for reasons unrelated to objective quality, could the authors clarify how these noisy signals are filtered or how the model can learn a clear preference signal from this noisy data?
>
> **A1:** This is an excellent question that points to the core insight of our work: **models can learn unified and general preference representations from noisy human preference signals**, and we argue that this "noise" is actually a necessary aspect of general preference.
>
> We distinguish between human preference "noise" and traditional data noise: Even when users upvote posts for subjective rather than objective reasons, these remain **real human preferences**. These preferences may be individual or idiosyncratic, but they constitute fragments of the world's human preferences. If we impose rules to filter out these fragments, we would only learn the preferences we prescribed, rather than truly universal, world-scale human preferences.
>
> In the early stages of our research, we also believed that noise in preference data would impair preference learning, and we conducted extensive denoising. However, we found that data denoising typically only introduces inductive bias. While this may yield good short-term results on benchmarks, it ultimately harms generalization. We believe **the lesson of "The Bitter Lesson" applies here as well** [1]. Human preferences are extremely complex, and we should stop trying to prescribe them based on human knowledge (e.g., labeling genuine human preferences as "noise" and removing them). Instead, we should let AI learn the specifics of these preferences from massive data.
>
> **How do models learn clear preference signals from noisy data?** We can draw an analogy to language modeling. Next-token prediction also provides noisy signals—why does this task elicit powerful language generalization capabilities? A simple understanding is that **models cannot memorize so many noisy signals, and therefore must discover deep representations and intrinsic patterns in language**.
>
> Similarly, in preference modeling, humans provide noisy but genuine signals. Users may upvote for various reasons, and models likewise cannot simply memorize these patterns. Instead, **models must discover decisive, generalizable preference representations across numerous data points**. Only sufficiently universal preference features (e.g., objective quality) can capture more training samples, while less universal features (e.g., style) can only capture a subset of training data. **To fit the training data, models must discard idiosyncratic preferences and discover generalizable ones.**
>
> **Empirical evidence:** This mechanism is concretely demonstrated in Section 3.3.2 through the style dimension analysis. Preferring longer responses enables good fitting on 60% of the data but performs poorly on the remaining 40%. To capture more samples, the model must abandon this feature and discover more fundamental preference patterns.
>
> ---
>
>
>
> > **Q2:** It is somewhat difficult to assess the technical contribution of this work. While identifying scaling laws in preference modeling is valuable, the paper does not clearly articulate the specific technical challenges that WorldPM addresses beyond the empirical observation of scaling behavior.
>
> **A2:** We  would like to clarify the technical contributions and challenges addressed by our work from multiple perspectives:
>
> **1. Addressing a fundamental open question in preference modeling:**
>
> Scaling human preference modeling and building general preference/reward models is itself a key and challenging problem in the preference modeling field. We make the **novel and non-trivial claim that human preference modeling is scalable**, paving the way for constructing general reward models. This claim has not been explored or validated in prior work, and **it is not obviously true**. Simply assuming preference modeling is scalable would overlook the complexity of this challenge. Our experiments reveal this complexity through several findings: not all model sizes exhibit scaling on objective tasks, and no clear scaling is observed on subjective tasks.
>
> These phenomena reveal both the **difficulty of constructing general reward models** and the **complexity of properly evaluating them**. As reviewer APCU noted: "In terms of significance, I believe many groups will consult this paper when deciding whether to invest in PM data vs. scaling the base model vs. changing evaluation protocols" and reviewer DUmw noted "The motivation of the paper is strong and timely, as building a general-purpose preference model is a natural next step in advancing alignment research."

---

> ### Author Response · Authors · 2025-11-24
> **Response to Reviewer (2/3)**
>
> **2. Solving preference learning under weak supervision:**
>
> Your first question—"how the model can learn a clear preference signal from noisy forum data"—represents a concern that many readers likely share. This highlights another important perspective on our contribution: **we address the problem of learning preference representations under weak supervision**.
>
> We demonstrate that **noisy, diverse human preferences can be successfully aggregated by models to generate generalizable human preference representations**—this is precisely why we propose the concept of "World Preference Modeling." This perspective has been **overlooked by all prior work**, which typically treats human preference noise as something to be filtered out through denoising, thus hindering consideration of whether preference modeling is scalable and whether generalizable preference representations can be learned.
>
> **In summary**, we believe our work makes important contributions to the development of human preference modeling by:
>
> - Establishing the scalability of preference modeling with rigorous empirical evidence
> - Identifying the conditions under which scaling occurs (model capacity thresholds, evaluation dimensions)
> - Demonstrating that generalizable preferences can be learned from weakly supervised, noisy data
> - Providing practical guidance for building general reward models
> - providing a strong initialization for preference fine-tuning to facilitate community research
>
> ---
>
>
>
> > **Q3:** The notation $Z$ in Equation (3), representing the style difference, appears to be inconsistently defined. In L249, the authors state that $Z\in R^S$ is the style features vector. In Equation (3), how to derive the style features vectors $Z$ for responses $y_0$ and $y_1$?
>
> **A3:** We thank the reviewer for pointing out this notational inconsistency. We have revised the notation to clearly distinguish between the style feature vector and the style difference:
>
> - **$V$** denotes the style feature vector
> - **$Z$** denotes the normalized style difference computed from $V$
>
> The revised Equation (3) now clearly defines the element-wise computation:
>
> $$
> Z_i(x, y_0, y_1) = \text{normalize}\left(\frac{V_i(y_0) - V_i(y_1)}{V_i(y_0) + V_i(y_1)}\right), \quad i = 1,\ldots,S
> $$
>
> where $V_i(y)$ represents the $i$-th element of the style feature vector for response $y$, and $S$ is the dimensionality of the style feature space. In our experiments, we use $V = [\text{token length}, \text{markdown lists count, headers count, bold text count}]$, which can be extracted from any response $y$. These features (token length and markdown formatting) are well-known metrics that are prominently exploited during LLM optimization and reward hacking.
>
> This clarification has been added to Section 3.3.2 and Equation 3 caption.

---

> ### Author Response · Authors · 2025-11-24
> **Response to Reviewer (3/3)**
>
> > **Q4:** The evaluation of stylistic features is limited to four attributes (token length, markdown lists, headers, and bold text), overlooking potentially important stylistic dimensions such as syntactic complexity.
>
> **A4:** The objective of this evaluation is to explain the potential reasons why subjective evaluations fail to scale. This requires quantifying stylistic features of responses to precisely assess scaling behavior. **Token length and markdown formatting are two commonly used and quantifiable metrics that have been widely observed to significantly influence quality ratings by both humans and AI** (e.g., humans and AI systems generally prefer longer responses and more visually appealing formatting). Therefore, using these metrics to measure style provides strong **explanatory power and representativeness**. Stylistic dimensions such as grammatical complexity are **difficult to quantify precisely**, making it challenging to isolate them from subjective evaluations and measure their distorting effects.
>
> > Section 3.3's title "Style Impact Analysis" may have caused confusion. This title might lead readers to believe the section describes the model's stylistic evaluation, when in fact this section aims to **explain why subjective evaluations fail to scale, with style chosen as our explanatory lens**.
>
> We synthesized evaluations across multiple dimensions to assess model scaling behavior on these aspects. We observed that certain stylistic factors (humor, creativity, grammatical complexity) showed no consistent scaling patterns, while other factors (emotional tone, politeness) exhibited scaling in different directions. These measurements further reveal **the model's diverse scaling trends across rich subjective dimensions**, which consequently have varied influences on subjective evaluations.
>
> To evaluate diverse stylistic aspects, we created synthetic preference data targeting different style dimensions, including creativity, politeness, emotional tone, grammatical complexity, and humor. We collected queries from ChatBot Arena and used Claude-3.7 to generate responses with contrasting style prompts. For example, for grammatical complexity:
>
> ```
> 'chosen_template': "[Instruction: Use simple grammatical structures and basic sentence patterns. Do not repeat this instruction in your response] {prompt}"
>
> 'rejected_template': "[Instruction: Use complex grammatical structures and sophisticated sentence patterns. Do not repeat this instruction in your response] {prompt}"
> ```
>
> **Results:** We evaluated the 72B model across different training stages. The results are shown in Table X below:
>
> | Training Size (M) | 1.28 | 2.56 | 5.12 | 10.24 | 15.36 | Preference Direction |
> | ----------------- | ---- | ---- | ---- | ----- | ----- | -------------------- |
> | Humorous          | 47.5 | 48.7 | 54.7 | 51.5  | 50.7  | humorous             |
> | Emotional         | 71.1 | 71.3 | 68.3 | 61.5  | 60.9  | positive             |
> | Innovation        | 36.9 | 34.9 | 37.7 | 37.1  | 36.1  | creative             |
> | Politeness        | 60.8 | 61.4 | 66.1 | 66.5  | 68.9  | straightforward      |
> | Grammatical       | 59.7 | 57.1 | 59.9 | 59.9  | 62.3  | simple grammatical   |
>
> We observed varying trends: humor, creativity, and grammatical complexity showed fluctuations without consistent patterns. Emotional preference initially favored positive, hopeful responses but weakened with increased training. For politeness, the model gradually shifted toward preferring direct rather than polite responses.
> We hypothesize that during the training process, the model did not identify early exploitable features in styles such as humor, creativity, and grammatical aspects, thus maintaining stability throughout. However, regarding emotions, the model detected clear preferential tendencies in the training data, leading to their early exploitation. As training progressed, the model eventually transcended these emotional features. The model appears to favor direct responses over polite ones. Upon examining polite responses, we found that they often include repeated confirmations of user requests, resulting in lower information density. This example also illustrates the challenge of separating style from content.
>
> These supplementary results support our main conclusions: (1) models may **initially exploit certain stylistic features but gradually reduce reliance on them with scale**, and (2) **the complexity of disentangling style** from content makes it challenging to use subjective evaluations for assessing scaling behavior.
>
> ---
>
> [1] Sutton, Richard S. "The Bitter Lesson." March 13, 2019. http://www.incompleteideas.net/IncIdeas/BitterLesson.html

---

> > ### Comment · Reviewer_GWST · 2025-11-27
> >
> > I thank the authors for their clarification and response. After considering the authors’ rebuttal and reviewing the comments of the other reviewers, I still find it difficult to assess the technical contribution of the work. Although I acknowledge that the authors scale the human preference modeling through (1) large scale data collection and (2) extensive computational resources, I agree with the assessment of `Reviewer kEFN` that *the paper’s positioning and methodological rigor remain insufficiently resolved*. Therefore, I will keep my final score.

---

> > > ### Author Response · Authors · 2025-11-28
> > > **Response to Reviewer**
> > >
> > > We would appreciate further clarification on what constitutes "technological innovation" in this context, as it would help us better address your concerns. To the best of our knowledge, the scaling laws of preference modeling have not been explored in prior work, and we believe this represents a promising direction toward achieving general human preference modeling. We would be grateful if you could help us understand your perspective on why this contribution may not meet the criteria for technological innovation.

---

### Official Review · Reviewer_APCU · 2025-11-01

**Soundness:** 3
**Presentation:** 3
**Contribution:** 3
**Rating:** 6
**Confidence:** 4

**Summary:**

The paper proposes WorldPM, a large-scale preference‑model (PM) pre‑training setup built from forum‑sourced preference pairs (primarily StackExchange) and uses it to study how PM performance scales with both data size (up to ~15M pairs) and base model size (Qwen2.5 family, ~1.5B→72B). The authors report a three‑regime picture: (i) on adversarial PM evaluations, error decreases smoothly with more data/model size and admits power‑law fits; (ii) on objective, exactly‑graded tasks, improvements appear to be “emergent,” materializing only for larger base models; and (iii) on subjective, LLM‑judge benchmarks, headline scores do not cleanly improve with scale, which they attribute to style biases (e.g., response length, Markdown). They also introduce a simple “style‑control” adjustment that linearly regresses judge preferences on style features and an RM score difference, and they show that initializing downstream PM fine‑tuning with WorldPM yields consistent gains on several benchmarks.

**Strengths:**

I think this is a timely and useful empirical study. The originality is not a new algorithm but a comprehensive, carefully run scaling investigation specifically for PM pre‑training. That niche—reward/preference models rather than policies—has fewer systematic, large‑sweep studies, so mapping it out has value.

On quality, the experimental breadth is strong: multiple model sizes and data scales and a diverse evaluation suite spanning adversarial, objective, and subjective categories. The three‑regime picture is practically meaningful, and it affects how one might allocate PM compute and data depending on the target use. I also appreciated the diagnosis of style bias in LLM‑judge evaluations and the attempt—however imperfect—to factor it out. The downstream transfer result (WorldPM → better fine‑tuned PMs on objective tasks) is a concrete takeaway that practitioners can use.

Clarity is generally good: figures make the high‑level trends easy to grasp, and the paper is upfront about evaluation choices and their rationale. In terms of significance, I believe many groups will consult this paper when deciding whether to invest in PM data vs. scaling the base model vs. changing evaluation protocols; that makes it a useful reference point even if some conclusions confirm community intuitions.

**Weaknesses:**

1) Positioning and novelty claim need tightening (and up‑to‑date citations).
Several headline messages echo things the community already suspected: “more/bigger data helps,” “LLM‑judge metrics are style‑sensitive,” and “objective metrics tend to be more robust than subjective ones.” The value of the paper is the scope and quantification, not the qualitative direction. That’s fine—but the Related Work should be sharper and current. As submitted, I did not see a single 2025 reference, which is not acceptable for a September‑2025 submission in a fast‑moving area. The paper should explicitly situate itself against closely adjacent lines (reward‑overoptimization/scaling in RLHF on the policy side; recent generative reward models/RLAIF‑style RMs; 2025 analyses of LLM‑as‑judge biases and debiasing). A small related‑work matrix contrasting what scales (pairs vs. compute vs. base capacity), data provenance (human/AI/forum), and evaluation types would help calibrate what is genuinely new here.

2) Style‑control methodology risks leakage/over‑correction.
The linear adjustment re‑learns the weight on the RM score difference together with style features on the evaluation distribution. Without cross‑fitting or a held‑out split for learning the correction, the adjusted metric can absorb peculiarities of the test set and artificially “fix” rankings. I strongly recommend cross‑fit evaluation (learn the adjustment on split A, report on split B) and reporting both the raw and cross‑fit‑controlled results, plus a sensitivity analysis showing whether system rankings actually stabilize.

3) Some core results feel “obvious,” and the paper should own that and sharpen the non‑obvious parts.
I don’t want to undermine the work—running these experiments well is non‑trivial—but parts of the story are indeed expected (bigger PMs + more data → better adversarial outcomes; subjective metrics correlate with length/format). I suggest reframing the contribution more explicitly as measuring the shape and turning points of these curves, and highlighting what is truly non‑obvious (e.g., the “objective emergence” only beyond a base‑model size threshold; the apparent stability of power‑law exponents across tasks).

5) Batch‑size ablation doesn’t isolate the variable of interest.
Conclusions about “larger batch helps” are drawn under a protocol where total tokens increase with batch (fixed steps). That largely measures “more data helps.” Please re‑run with matched total tokens (or matched FLOPs/compute) to support any claim about batch itself.

6) Power‑law fits need better statistical treatment.
Power‑law exponents are reported over averages of a few benchmarks, with limited goodness‑of‑fit diagnostics. Per‑task fits with confidence intervals, checks for sensitivity to task inclusion, and compute‑normalized views would make the “scaling law” claims more robust.

8) Notation and mathematical correctness issues that need fixing.
I’m flagging three concrete items that should be corrected or clarified:
* I don't like the use of $D^\top \alpha$ when $D$ is introduced as a scalar. It might be a matter of personal preference, but I would rather see $ D \alpha$ in this case.
* In Equation 3, a fraction is written with $Z$ in the numerator and denominator while $Z$ is defined as a vector. As written, the expression is not well‑typed. Please clarify (for example if it is meant to be an element-wise operation) or use better notations.
* In the appendix illustrating chosen vs. rejected responses, the color theme appears inverted (green for rejected, red for chosen) for some of the examples. Please swap or label more clearly; this kind of visual miscue invites reader error.

**Questions:**

Every item raised in the Weaknesses section can be viewed as a question for the authors.
I may well be mistaken on several of these points, and I would sincerely appreciate clarification or correction wherever appropriate.
If the authors can address or resolve even part of these concerns—whether by showing that I misunderstood something or by providing additional detail—it would be very helpful.

**Details Of Ethics Concerns:**

* The “WorldPM” preferences are mined from forums (primarily StackExchange). That population is not demographically representative, and the paper does not include a dedicated bias audit. There’s a real risk that forum‑specific preferences (tone, dialect, topical norms) get encoded and amplified by the PM, which can systematically disadvantage certain groups or styles.
* Even with public data, posts can contain personal information or quasi‑identifiers. Re‑identification or unintended exposure is possible if raw text or URLs are included. An ethics review can verify the adequacy of de‑identification, access controls, and redaction policies in the planned release.
* Although votes/acceptance come from public forums (not paid annotators), releasing large‑scale preference datasets still raises documentation, consent, and provenance questions (e.g., how acceptance votes were converted to labels, what filtering was applied, whether users can request removal). Clear data cards and opt‑out mechanisms would be prudent.

---

> ### Author Response · Authors · 2025-11-24
> **Response to Reviewer (1/4)**
>
> We sincerely thank you for the comprehensive and constructive feedback. We appreciate the recognition of our work as "a timely and useful empirical study" with "strong experimental breadth" and "practically meaningful" findings. We have carefully addressed each concern and made substantial revisions to the manuscript. Below are our point-by-point responses.
>
>
> > **Q1:** Positioning and novelty claim need tightening (and up‑to‑date citations). Several headline messages echo things the community already suspected: "more/bigger data helps," "LLM‑judge metrics are style‑sensitive," and "objective metrics tend to be more robust than subjective ones." The value of the paper is the scope and quantification, not the qualitative direction. That's fine—but the Related Work should be sharper and current. As submitted, I did not see a single 2025 reference, which is not acceptable for a September‑2025 submission in a fast‑moving area. The paper should explicitly situate itself against closely adjacent lines (reward‑overoptimization/scaling in RLHF on the policy side; recent generative reward models/RLAIF‑style RMs; 2025 analyses of LLM‑as‑judge biases and debiasing). A small related‑work matrix contrasting what scales (pairs vs. compute vs. base capacity), data provenance (human/AI/forum), and evaluation types would help calibrate what is genuinely new here.
>
> **A1:** We thank you for this important suggestion. We have substantially revised and expanded the Related Work section to provide sharper positioning and include up-to-date citations. The revised content is as follows:
>
> Existing preference modeling and alignment methods can be primarily divided into two categories: The first category is scalar reward modeling based on the Bradley-Terry (BT) model. The second category involves text-based generative reward modeling methods like GenRM and LLM-as-judge [1,2,3,4]. WorldPM focuses on the BT scalar model and investigates its scalability. Previous research often utilized small-scale, high-quality human-annotated datasets (typically containing $10^3$--$10^5$ examples) [5,6] or larger AI-annotated datasets that may exhibit systematic biases [7,8]. In contrast, WorldPM leverages large-scale forum data, which inherently contains human preference signals to empirically study scaling laws in reward modeling.
>
> Another line of research examines the vulnerabilities of reward models during policy optimization and evaluation. With a fixed reward model, enhanced optimization often leads to overoptimization and Goodhart-like effects [9]. Additionally, reward models and LLM-as-judge systems exhibit biases in verbosity, self-preference, and adversarial settings [10,11], prompting research into debiasing and calibration techniques [12,13,14,15]. These studies typically fix the reward model or evaluator and adjust only optimization procedures or inputs. In contrast, WorldPM demonstrates that expanding the training scale can enhance objective discrimination and reduce biases.
>
> We have updated the Related Work section (Section 2.2) with this revised structure and added relevant 2025 references to position our work within the current research landscape.
>
> ---
>
> > **Q2:** Style‑control methodology risks leakage/over‑correction. The linear adjustment re‑learns the weight on the RM score difference together with style features on the evaluation distribution. Without cross‑fitting or a held‑out split for learning the correction, the adjusted metric can absorb peculiarities of the test set and artificially "fix" rankings. I strongly recommend cross‑fit evaluation (learn the adjustment on split A, report on split B) and reporting both the raw and cross‑fit‑controlled results, plus a sensitivity analysis showing whether system rankings actually stabilize.
>
> **A2:** We have implemented the cross-fit evaluation protocol as recommended:
>
> **Methodology:** We randomly split the dataset into Split A and Split B, train the style-control adjustment on Split A, test and report results on Split B. We performed multiple random splits and report standard deviations to ensure robustness.
>
> **Results:** We have updated the results in the main text (Figure 2, Page 5). The cross-fit evaluation results are **consistent and stable with the original results**: there are substantial differences before and after style control, and these differences gradually expand as training progresses. This confirms that our style-control methodology is not overfitting to test set peculiarities but capturing genuine style-related variance.

---

> ### Author Response · Authors · 2025-11-24
> **Response to Reviewer (2/4)**
>
> > **Q3:** Some core results feel "obvious," and the paper should own that and sharpen the non‑obvious parts. I don't want to undermine the work—running these experiments well is non‑trivial—but parts of the story are indeed expected (bigger PMs + more data → better adversarial outcomes; subjective metrics correlate with length/format). I suggest reframing the contribution more explicitly as measuring the shape and turning points of these curves, and highlighting what is truly non‑obvious (e.g., the "objective emergence" only beyond a base‑model size threshold; the apparent stability of power‑law exponents across tasks).
>
> **A3:** **Regarding adversarial evaluation:** We want to ensure our definition of "adversarial" is clearly understood, **as "adversarial" has a distinct meaning in PM evaluation compared to its conventional usage**. In this paper, "adversarial" refers to a specific subclass commonly found in community RM benchmarks, such as OffsetBias, the chat subclass in RMBench, and the chat-hard subclass in RewardBench. These evaluations typically construct preference pairs consisting of a normal response and a response with factual errors/subtle defects/biases. This differs from traditional robustness evaluation in that both responses are fluent and realistic, whereas traditional robustness is generally achieved by adding perturbations to the input. The evaluation tests whether preference models can identify factual errors, subtle defects, and biases. **To our knowledge, this finding has not been mentioned in prior RM/PM research**, making it a non-obvious contribution.
>
> **Regarding subjective evaluation:** Our research on subjective aspects aims to explain why subjective evaluations may not reflect scalability. **We mention that subjective evaluations are correlated with style, but this is not our primary conclusion.** We show that (Section 3.3.1, 3.3.2):
>
> (1) Models rely on style features for judgments in early training, but this bias gradually diminishes with continued scaling, and we explain how models overcome stylistic bias.
>
> (2) As model bias weakens, results become more divergent on biased subjective evaluations. Therefore, we infer that subjective evaluations, due to mixing various dimensions of evaluation and conflicts in certain dimensions (e.g., style), **cannot actually reflect the model's scaling performance**.
>
> We hope these explanations clarify our contributions. Please let us know if we have misunderstood any of your points.
>
> ---
>
> > **Q4:** Batch‑size ablation doesn't isolate the variable of interest. Conclusions about "larger batch helps" are drawn under a protocol where total tokens increase with batch (fixed steps). That largely measures "more data helps." Please re‑run with matched total tokens (or matched FLOPs/compute) to support any claim about batch itself.
>
> **A4:** We want to clarify our experimental design and have added the requested matched-token comparison:
>
> **Our original intention:** When conducting this experiment, we wanted to demonstrate a different perspective, which is why we adopted this setting. We believe preference modeling pretraining is a task with substantial noise, and we wanted to investigate **how many samples are needed per gradient update to adequately estimate gradients or compress noise**. Our implicit assumption is that there exists a performance-saturating batch size, beyond which each update step brings similar or harmful optimization.
>
> **Matched-token comparison:** We have updated the main text with performance comparisons under the same training volume (10M samples), as shown in **Figure 9**. The experiments demonstrate that **under the same training volume, within the range of 2.5K~40K batch sizes, there is still a trend of better performance with larger batch sizes**.
>
>
> ---
>
> > **Q5:** Power‑law fits need better statistical treatment. Power‑law exponents are reported over averages of a few benchmarks, with limited goodness‑of‑fit diagnostics. Per‑task fits with confidence intervals, checks for sensitivity to task inclusion, and compute‑normalized views would make the "scaling law" claims more robust.
>
> **A5:** We thank you for this suggestion to strengthen our statistical analysis. We have added the requested analyses:
>
> **(1) Per-task fits with goodness-of-fit:** We report individual task-level scaling law fits with corresponding goodness-of-fit metrics (R² values) in Figure 14, demonstrating that the power-law relationship holds consistently across different evaluation tasks. Below, the values in parentheses correspond to $R^2$, where tasks with $R^2 \geq 0.9$ are shown in bold, indicating strong power-law fits.

---

> ### Author Response · Authors · 2025-11-24
> **Response to Reviewer (3/4)**
>
> | Task                              | 1.5B                                    | 7B                                      | 32B                                     | 72B                                     |
> | --------------------------------- | --------------------------------------- | --------------------------------------- | --------------------------------------- | --------------------------------------- |
> | **Factual Error Detection** | $L=-0.273D^{-0.129}$**(0.998)** | $L=-0.404D^{-0.148}$**(0.991)** | $L=-0.515D^{-0.106}$**(0.983)** | $L=-0.600D^{-0.117}$**(0.974)** |
> | **OffsetBias**              | $L=-0.308D^{-0.129}$**(0.991)** | $L=-0.701D^{-0.093}$**(0.975)** | $L=-0.664D^{-0.132}$**(0.986)** | $L=-0.773D^{-0.091}$**(0.902)** |
> | **LLMBar**                  | $L=-0.239D^{-0.061}$(0.882)           | $L=-0.417D^{-0.086}$**(0.944)** | $L=-0.449D^{-0.073}$**(0.919)** | $L=-0.548D^{-0.056}$(0.842)           |
> | **MBPP-Plus**               | $L=-0.373D^{-0.005}$(0.673)           | $L=-0.491D^{-0.022}$(0.711)           | $L=-0.496D^{-0.040}$**(0.911)** | $L=-0.507D^{-0.021}$**(0.903)** |
> | **MMLU-Pro**                | $L=-0.371D^{-0.001}$(0.027)           | $L=-0.449D^{-0.017}$(0.715)           | $L=-0.466D^{-0.038}$**(0.957)** | $L=-0.513D^{-0.033}$**(0.897)** |
> | **IFEval**                  | $L=-0.349D^{-0.001}$(0.113)           | $L=-0.367D^{-0.003}$(0.633)           | $L=-0.369D^{-0.004}$(0.891)           | $L=-0.384D^{-0.011}$**(0.957)** |
> | **HumanEvalPack**           | $L=-0.362D^{-0.005}$(0.493)           | $L=-0.390D^{0.003}$(0.310)            | $L=-0.400D^{-0.017}$**(0.968)** | $L=-0.419D^{-0.019}$**(0.959)** |
> | **GPQA**                    | $L=-0.372D^{0.000}$(0.014)            | $L=-0.370D^{0.004}$(0.706)            | $L=-0.380D^{-0.002}$(0.208)           | $L=-0.384D^{-0.007}$**(0.962)** |
> | **MATH**                    | $L=-0.365D^{-0.005}$(0.389)           | $L=-0.424D^{-0.041}$**(0.923)** | $L=-0.492D^{-0.056}$**(0.966)** | $L=-0.469D^{-0.033}$**(0.901)** |
> | **Chatbot Arena**           | $L=-0.387D^{0.009}$(0.803)            | $L=-0.435D^{0.002}$(0.068)            | $L=-0.425D^{-0.003}$(0.462)           | $L=-0.441D^{0.004}$(0.400)            |
> | **HelpSteer2**              | $L=-0.388D^{0.002}$(0.153)            | $L=-0.493D^{-0.011}$(0.453)           | $L=-0.475D^{-0.013}$(0.748)           | $L=-0.529D^{-0.005}$(0.224)           |
> | **GPT4**                    | $L=-0.464D^{0.027}$(0.709)            | $L=-0.638D^{0.009}$(0.066)            | $L=-0.586D^{-0.004}$(0.149)           | $L=-0.594D^{0.002}$(0.018)            |
>
> **(2) Compute-normalized views:** We have added **Figure 15**, which presents the scaling curves of each task from a compute-normalized perspective, showing that our conclusions remain robust when viewed through the lens of computational resources.
>
> ---
>
> > **W6:** Notation and mathematical correctness issues that need fixing.
> >
> > - I don't like the use of $D^T α$ when $D$ is introduced as a scalar. It might be a matter of personal preference, but I would rather see $D α$ in this case.
> > - In Equation 3, a fraction is written with $Z$ in the numerator and denominator while $Z$ is defined as a vector. As written, the expression is not well‑typed. Please clarify (for example if it is meant to be an element-wise operation) or use better notations.
> > - In the appendix illustrating chosen vs. rejected responses, the color theme appears inverted (green for rejected, red for chosen) for some of the examples. Please swap or label more clearly; this kind of visual miscue invites reader error.
>
> **A6:**
>
> We thank the reviewer for catching these notation issues. We have made the following corrections:
>
> **(1) Notation for $D^T α$:** We have changed the notation from $D^T α$ to $D α$ as suggested.
>
> **(2) Equation 3 clarification:** We have clarified that the operation is element-wise. The revised equation now reads:
>
> $$
> Z_i(x, y_0, y_1) = \text{normalize}\left(\frac{V_i(y_0) - V_i(y_1)}{V_i(y_0) + V_i(y_1)}\right), \quad i = 1,\ldots,S
> $$
>
> where $V$ denotes the style feature vector and $Z$ represents the style difference.
>
> **(3) Color theme in appendix:** We appreciate the concern about visual clarity. However, **the inverted color scheme has special meaning**: the examples we list are actually **real data from the test set with incorrect labels**. We use inverted colors deliberately to highlight these label errors. (e.g. Figure 20 is an example from the PPE-Human test set where the chosen response's proof is wrong while the rejected response's proof is correct, and the WorldPM identify this error.)

---

> ### Author Response · Authors · 2025-11-24
> **Response to Reviewer (4/4)**
>
> ### Responses to Ethics Concerns
>
> We appreciate the reviewer's attention to ethical considerations. We had already anticipated these concerns and provided detailed explanations in the **Ethics Statement**. Below we address each specific concern:
>
> > **Ethics Concern 1:** The "WorldPM" preferences are mined from forums (primarily StackExchange). That population is not demographically representative, and the paper does not include a dedicated bias audit. There's a real risk that forum‑specific preferences (tone, dialect, topical norms) get encoded and amplified by the PM, which can systematically disadvantage certain groups or styles.
>
> **Response:** We acknowledge that forum-based data may contain specific preferences. Therefore, in our Ethics Statement (Page 10), **we recommend not directly deploying WorldPM, but rather fine-tuning the model based on WorldPM**. This approach can mitigate group-specific or stylistic biases while leveraging the generalizable preference knowledge learned during pretraining.
>
> > **Ethics Concern 2:** Even with public data, posts can contain personal information or quasi‑identifiers. Re‑identification or unintended exposure is possible if raw text or URLs are included. An ethics review can verify the adequacy of de‑identification, access controls, and redaction policies in the planned release.
>
> **Response:** **We plan to release the model rather than the dataset**, as the dataset is publicly available and anyone can generate our dataset following the methods described in this paper. Furthermore, during training, **we removed personal information and quasi-identifiers to prevent user privacy leakage**.
>
> > **Ethics Concern 3:** Although votes/acceptance come from public forums (not paid annotators), releasing large‑scale preference datasets still raises documentation, consent, and provenance questions (e.g., how acceptance votes were converted to labels, what filtering was applied, whether users can request removal). Clear data cards and opt‑out mechanisms would be prudent.
>
> **Response:** We provide detailed documentation of our data sources in the **Data Licensing and Usage** section. StackExchange provides public data dumps that anyone can access under the CC BY-SA 4.0 license. Sections 3.1 and A provide comprehensive explanations of our data collection process and detailed data analysis.
> Since we are releasing models rather than data, and the source data is already publicly available under permissive licenses, users who wish to be excluded from the original forums can follow StackExchange's existing opt-out mechanisms.
>
> ---
>
> [1] Findeis, Arduin, et al. "Inverse Constitutional AI: Compressing Preferences into Principles." ArXiv, vol. abs/2406.06560, 2024.
>
> [2] Liu, Zijun, et al. "Inference-Time Scaling for Generalist Reward Modeling." arXiv preprint arXiv:2504.02495, 2025.
>
> [3] Chen, Xiusi, et al. "RM-R1: Reward Modeling as Reasoning." arXiv preprint arXiv:2505.02387, 2025.
>
> [4] Gu, Jiawei, et al. "A Survey on LLM-as-a-Judge." arXiv preprint arXiv:2411.15594, 2025.
>
> [5] Wang, Zhilin, et al. "HelpSteer3-Preference: Open Human-Annotated Preference Data across Diverse Tasks and Languages." arXiv preprint arXiv:2505.11475, 2025.
>
> [6] Cui, Ganqu, et al. "Ultrafeedback: Boosting Language Models with High-Quality Feedback." arXiv preprint arXiv:2310.01377, 2023.
>
> [7] Chiang, Cheng-Han, and Hung-yi Lee. "Can Large Language Models Be an Alternative to Human Evaluations?" arXiv preprint arXiv:2305.01937, 2023.
>
> [8] Fu, Xiyan, and Wei Liu. "How Reliable is Multilingual LLM-as-a-Judge?" arXiv preprint arXiv:2505.12201, 2025.
>
> [9] Gao, Leo, John Schulman, and Jacob Hilton. "Scaling Laws for Reward Model Overoptimization." Proceedings of the International Conference on Machine Learning, PMLR, 2023, pp. 10835-10866.
>
> [10] Spiliopoulou, Evangelia, et al. "Play Favorites: A Statistical Method to Measure Self-Bias in LLM-as-a-Judge." ArXiv, vol. abs/2508.06709, 2025.
>
> [11] Chen, Guiming Hardy, et al. "Humans or LLMs as the Judge? A Study on Judgement Biases." arXiv preprint arXiv:2402.10669, 2024.
>
> [12] Choi, Hyeong Kyu, Xiaojin Zhu, and Yixuan Li. "Measuring and Mitigating Identity Bias in Multi-Agent Debate via Anonymization." ArXiv, vol. abs/2510.07517, 2025.
>
> [13] Satterfield, Reed, et al. "One Token to Fool LLM-as-a-Judge." arXiv preprint arXiv:2507.08794, 2025.
>
> [14] Dai, Juntao, et al. "Mitigating Reward Over-Optimization in RLHF via Behavior-Supported Regularization." ArXiv, vol. abs/2503.18130, 2025.
>
> [15] Wolf, Lorenz, Robert Kirk, and Mirco Musolesi. "Reward Model Overoptimisation in Iterated RLHF." arXiv preprint arXiv:2505.18126, 2025.

---

### Official Review · Reviewer_DUmw · 2025-11-01

**Soundness:** 3
**Presentation:** 3
**Contribution:** 3
**Rating:** 4
**Confidence:** 4

**Summary:**

This paper introduces World Preference Modeling (WorldPM), a large-scale pretrained preference model trained on hundreds of billions of preference-aligned samples across multiple domains. The authors argue that large-scale pretraining enables the model to acquire broad and generalizable preference knowledge, thereby simplifying downstream preference learning and improving generalization. Overall, the work presents an impressive large-scale effort that aims to lay the foundation for scalable preference modeling.

**Strengths:**

1.The motivation of the paper is strong and timely, as building a general-purpose preference model is a natural next step in advancing alignment research.

2.The authors have conducted a technically demanding and large-scale project, involving extensive data collection, training, and evaluation.

3.The results demonstrate that large-scale preference pretraining can improve model generalization, reduce stylistic bias, and accelerate downstream alignment tasks.

**Weaknesses:**

1.Ambiguity remains in defining what World Preference Learning actually covers, leaving unclear whether it includes physical, visual, auditory, or embodied modalities.

2.Insufficient clarification is provided for the adversarial setting, which can have different meanings across domains and therefore needs a more precise explanation.

3.Related work lacks a coherent structure and fails to connect this study with foundational methods such as RLHF, DPO, GRPO.

4.Analysis of results stays largely descriptive, with key claims—such as reducing stylistic bias through scale—unsupported by quantitative or causal evidence.

5.Potential reward hacking and overfitting to stylistic features are mentioned but never thoroughly examined, leaving uncertainty about what biases remain after pretraining.

6.Discussion of annotator bias and value alignment is missing, despite their central importance to subjective preference modeling.

7.Impact of contradictory preference labels on models of varying scales is unexplored, preventing insight into how model capacity mediates robustness under conflicting supervision.

**Questions:**

1.How is “World Preference Learning” formally defined, and does it extend to multimodal or embodied forms such as video, physical interaction, or speech?

2.What exactly does the term “adversarial” mean in the context of this paper—does it refer to adversarial prompting, preference conflict, or robustness evaluation—and how was this setting implemented in experiments?

3.Could the authors expand the related work section to include a clear taxonomy of previous preference modeling and alignment methods, highlighting how WorldPM differs from or builds upon prior frameworks such as DPO, GRPO?

4.The statement that models initially rely on stylistic features but reduce this bias with more data is interesting, but can this trend be quantitatively demonstrated, for example through feature attribution or bias disentanglement analysis?

5.If the data contains irrelevant or misleading features correlated with preference labels, does the model eventually unlearn these patterns with scale, or does large-scale pretraining risk amplifying such reward-hacking behavior?

6.For subjective datasets, how are annotators trained to ensure value consistency, and have the authors analyzed whether differences in cultural or reasoning styles influence the learned preference distribution?

7.Has the paper examined how models of different sizes handle contradictory preference data, and whether there is a threshold of conflict severity that causes accuracy degradation?

8.Would the proposed framework generalize to multimodal or embodied preference learning tasks, and if so, what adjustments would be required to extend it beyond text-based preferences?

---

> ### Author Response · Authors · 2025-11-24
> **Response to Reviewer (1/3)**
>
> > **W1/Q1:** Ambiguity remains in defining what World Preference Learning actually covers, leaving unclear whether it includes physical, visual, auditory, or embodied modalities. How is "World Preference Learning" formally defined, and does it extend to multimodal or embodied forms such as video, physical interaction, or speech?
>
> **A1:** World Preference Modeling differs from traditional human preference modeling (which typically relies on small-scale, carefully curated human preference data) by pursuing **large-scale modeling of generalizable human preferences**. This approach is grounded in two theoretical assumptions:
>
> 1. Despite substantial preference variations across individuals, humans share certain **general preference representations** (e.g., preferences for more truthful and objective responses).
> 2. These universal preference structures can be learned by AIs through **scaling of preference modeling**, where models do not cease learning despite variations in human preferences.
>
> From this perspective, if we accept that different human populations exhibit similar preference structures for visual and auditory modalities (e.g., shared understanding of good music across genres), World Preference Modeling should theoretically extend to other modalities. However, since current research and available data on human preferences remain predominantly text-focused, this paper limits its scope to the text domain.
>
> ---
>
> > **W2/Q2:** Insufficient clarification is provided for the adversarial setting, which can have different meanings across domains and therefore needs a more precise explanation. What exactly does the term "adversarial" mean in the context of this paper—does it refer to adversarial prompting, preference conflict, or robustness evaluation—and how was this setting implemented in experiments?
>
> **A2:** In this paper, "adversarial" refers specifically to robustness evaluation of preference models, which has a **domain-specific meaning in preference modeling**. It involves constructing preference pairs consisting of a normal response and a response with **subtle defects** to test whether preference/reward models can identify these nuanced flaws (i.e., good models should assign low scores to responses with subtle defects).
>
> Common methods for constructing such subtle defects include:
>
> - Inserting factual errors into responses
> - Creating superficially good but incomplete responses
>
> These adversarially constructed responses remain fluent and may even be imperceptible to humans, but they are adversarial in the context of preference modeling. This is distinct from adversarial prompting (attacks on input) or adversarial training (training robustness techniques) in other domains.
>
> Implementation in our experiments: We evaluate model robustness using established benchmarks that contain such adversarially constructed preference pairs, as detailed in Appendix B.2.
>
> ---
>
> > **W3/Q3:** Related work lacks a coherent structure and fails to connect this study with foundational methods such as RLHF, DPO, GRPO. Could the authors expand the related work section to include a clear taxonomy of previous preference modeling and alignment methods, highlighting how WorldPM differs from or builds upon prior frameworks?
>
> **A3:** We have revised and expanded the Related Work section to provide a clear taxonomy and better position our work. First, RLHF as an alignment framework consists of two stages: preference modeling and policy optimization. Methods such as DPO, GRPO, and PPO are all specific approaches for the policy optimization stage, while our work focuses on the preference modeling stage. Widely applied approaches in the preference modeling stage include Bradley-Terry (BT) model-based methods and text feedback-based methods, with the latter including approaches such as GenRM and LLM-as-a-judge[2,3,4]. Our work focuses on scaling up BT model-based preference modeling, investigating whether large-scale pretraining enables models to learn generalizable preference representations.

---

> ### Author Response · Authors · 2025-11-24
> **Response to Reviewer (2/3)**
>
> > **W4/Q4:** Analysis of results stays largely descriptive, with key claims—such as reducing stylistic bias through scale—unsupported by quantitative or causal evidence. Can this trend be quantitatively demonstrated, for example through feature attribution or bias disentanglement analysis?
>
> **A4:** We respectfully disagree with the assessment that our analysis lacks quantitative evidence. **Section 3.3.2 provides detailed quantitative experiments demonstrating this conclusion through two complementary analyses:**
>
> **Figure 3a** (Page 7): We quantitatively measure the **Phi coefficient between model predictions and stylistic features** (e.g., response length) across different training steps. The results show that this correlation **decreases progressively with training scale**  and converges toward the correlation between golden labels and stylistic features. This quantitatively demonstrates that models initially rely on stylistic features, but this stylistic bias diminishes with increased training scale.
>
> **Figure 3b** (Page 7): We perform stratified analysis by separating training samples into style-aligned and style-conflicting subsets. Training samples with **style-aligned preferences** (e.g., longer is better, which aligns with the overall correlation) achieve good fitting (high accuracy \~82%) in early training stages, while samples with **style-conflicting preferences** (e.g., longer is worse) initially show poor fitting (\~55% accuracy) but **continue to improve with training scale** (reaching ~75% accuracy). This quantitatively demonstrates that models initially rely on stylistic features for discrimination (resulting in poor fitting on style-conflicting data), but reduce this reliance with scale (enabling better prediction on style-conflicting data).
>
> ---
>
> > **W5/Q5:** Potential reward hacking and overfitting to stylistic features are mentioned but never thoroughly examined, leaving uncertainty about what biases remain after pretraining. If the data contains irrelevant or misleading features correlated with preference labels, does the model eventually unlearn these patterns with scale, or does large-scale pretraining risk amplifying such reward-hacking behavior?
>
> **A5:**
>
> Our experiments actually demonstrate that large-scale preference training progressively reduces biases. Please refer to the Adversarial evaluation, where OffsetBias is specifically a test set constructed to identify biases in reward models. On this test set, we observe continued scaling behavior. In terms of subjective preferences, we also show that stylistic biases are gradually diminishing. Therefore, within the scope of our observations, biases are consistently decreasing.
>
> We argue that irrelevant features in training data are **unlearned by models during large-scale training**. In fact, large-scale preference training more closely resembles unsupervised or weakly supervised learning, where the training data scale is sufficiently large that models cannot learn by simply memorizing the data. Therefore, models must seek sufficiently general and generalizable features. If the data contains misleading features (assuming these features appear in a minority of samples—if they appear in the majority, the data source itself may be unreliable), and the model exploits these misleading features, it will compromise generalization on training data that lacks these misleading features. Consequently, models may initially exploit these features to rapidly reduce training loss, but as training scales up, models must unlearn these features to achieve better generalization and lower training loss.
>
> **Empirical evidence from stylistic features:** Stylistic features can be viewed as a form of misleading features. As demonstrated in Section 3.3.2:
>
> - Models initially achieve rapid loss reduction by exploiting stylistic correlations
> - This exploitation leads to poor performance on style-conflicting samples (Figure 3b)
> - Continued scaling forces models to reduce reliance on stylistic features to improve overall performance
> - Final models show significantly reduced stylistic bias (Figure 3a)

---

> ### Author Response · Authors · 2025-11-24
> **Response to Reviewer (3/3)**
>
> > **W7/Q7:** Impact of contradictory preference labels on models of varying scales is unexplored, preventing insight into how model capacity mediates robustness under conflicting supervision. Has the paper examined how models of different sizes handle contradictory preference data, and whether there is a threshold of conflict severity that causes accuracy degradation?
>
> **A7:** Since our training data comes from forums, directly contradictory preference data (e.g., identical data with different labels) is unlikely. Instead, we observe preference conflicts, where some users prefer detailed responses while others prefer concise ones. However, we argue that **preference diversity differs from traditional data noise problems, as these preferences come from real humans and reflect the genuine complexity of human preferences**. Enforcing a single type of preference would introduce new biases.
>
> A prominent contribution of this paper is demonstrating that even when human preferences are rich and diverse, containing "noise," powerful models can still learn generalizable human preference representations from large-scale data (e.g., continued scaling on objectivity evaluation). Moreover, even small-capacity models (1B parameters) still generalize on robustness evaluations. There certainly exists a noise threshold (e.g., approaching 50%) beyond which models cannot learn meaningful preference correlations. However, our experiments actually impose extremely lenient requirements on training data—forum data involves no annotator instructions and relies entirely on spontaneous annotator behavior, representing a lower bound for scalable human preference learning.
>
> Our analysis of stylistic bias handling illuminates how models process conflicting preferences. For example, if 60% of training data prefers longer responses while 40% prefers shorter ones, this creates preference conflict. Models initially learn the "longer is better" feature because it enables rapid fitting on 60% of the data. However, to generalize on the remaining 40%, **models must unlearn or discard stylistic features and discover more unified and generalizable features beyond this conflict**. This can be viewed as how models progress from surface-level preferences to generalizable preference learning. Preference conflicts precisely reflect that such preferences are insufficiently universal and fundamental.
>
> ---
>
> > **Q8:** Would the proposed framework generalize to multimodal or embodied preference learning tasks, and if so, what adjustments would be required to extend it beyond text-based preferences?
>
> **A8:** The framework is theoretically modality-agnostic based on our assumptions. Our work demonstrates that human preferences are scalable in the text modality, making extensions to other modalities more promising. To extend to other modalities, the key requirements are: (1) collecting large-scale data from sufficiently diverse populations (e.g., through internet interactions), and (2) modeling preferences through supervised learning (in contrast, reinforcement learning may not be an effective "data sponge" for learning generalizable features from massive data [1]).
>
> ---
>
> [1] Eric Jang. "Generalization." Blog post, October 23, 2021. https://evjang.com/2021/10/23/generalization.html
>
> [2] Liu, Zijun, et al. "Inference-Time Scaling for Generalist Reward Modeling." arXiv preprint arXiv:2504.02495, 2025. https://arxiv.org/abs/2504.02495
>
> [3] Chen, Xiusi, et al. "RM-R1: Reward Modeling as Reasoning." arXiv preprint arXiv:2505.02387, 2025. https://arxiv.org/abs/2505.02387
>
> [4] Gu, Jiawei, et al. "A Survey on LLM-as-a-Judge." arXiv preprint arXiv:2411.15594, 2025. https://arxiv.org/abs/2411.15594

---

### Author Response · Authors · 2025-12-03
**Clarifying the Novelty and Contributions of Our Work**

We sincerely thank all reviewers and the area chair for their thoughtful reviews. During the discussion, we sense that the motivation and contributions of our work may have been misunderstood or underestimated.

The motivation of our work is to **build a general human preference model** (referred to as *world preference* in this paper). We believe the key obstacle to achieving this goal is that current preference modeling operates at a very limited scale, which restricts models to learning only partial inductive biases from small amounts of data. Therefore, we believe the path toward this goal lies in scaling human preference modeling—both in terms of data scale and diversity of human preference sources. Our research substantially exceeds prior work in both dimensions (involving millions of users and tens of millions of data points).

The most important contribution of our work is demonstrating that **human preference modeling is scalable, which establishes a viable path toward building general human preference models**. While there has been extensive research on scaling laws in other domains, it would be inappropriate to assume that the scalability of preference modeling is a natural corollary or "expected" finding. Human preference modeling is a fundamentally different research area, and to our knowledge, **no prior work has systematically investigated the scalability of preference modeling**. Moreover, our research reveals that preference modeling does not scale uniformly across all scenarios—this comprehensive characterization of such complexity represents our unique contribution.

This complexity manifests in two key aspects in our paper:

Base model size has a decisive impact on whether preference modeling scales. For example, 7B models do not exhibit scaling trends on many objective test sets. This emergent phenomenon indicates that preference modeling remains a highly challenging task, and certain discriminative capabilities absent in smaller models may emerge in larger ones.

Subjective evaluation sets are not suitable for testing scalability. We analyze the reasons: subjective evaluation compresses many dimensions, while models may not scale—or even scale inversely—along certain dimensions (e.g., style). This makes it difficult for subjective evaluation to reveal scalability. In contrast, objective and adversarial evaluations test single dimensions, allowing scaling behavior to manifest more clearly.

We believe Reviewer APCU provided a more objective assessment: "many groups will consult this paper when deciding whether to invest in PM data vs. scaling the base model vs. changing evaluation protocols; that makes it a useful reference point even if some conclusions confirm community intuitions."

---

### Meta-Review · Area_Chair_YSxq · 2026-01-16

**Summary:**

The authors propose "World Preference Modeling" which they claim is a "unified representation of human preferences". The models are trained on 15M-scale preference data that the authors collected from public forums, and range 1.5B to 72B. The paper studies how different metrics scale with these models. WorldPM is also put forward as a "foundation for preference fine-tuning".

I based my assessment on the following concerns, derived from the reviews:

W1. The related work section was found to be lacking by multiple reviewers, who mentioned that no recent references are included and that the papers fails to connect the contributions of this paper to existing foundation models
W2. The methodology was seen as prone to leakage and reward hacking
W3. Issues with the statistical soundness of the power-law fit
W4. Ambiguity concerning scope: the modalities the WoldPM covers, and in general its higher level goals and the methodological niche it fills
W5. Missing discussion of annotator bias and value alignment
W6. Claims ("reducing stylistic bias through scale", "larger batch helps") unsupported by evidence, and scaling results only reported for Qwen2.5 models
W7. Equation (3) was pointed as unclear by two of the reviewers.

**Reviewer Concerns:**

W1. The author responses attempted to address W1, though I still find the positioning of the paper with respect to related work unclear.
W2. Not adequately addressed.
W3. Addressed in answer to reviewer APCU response 3/4.
W4. The authors have addressed W4, in that it seems that only text is considered and provided further details on the higher-level goals of the work. I found the following illuminating: "humans share certain general preference representations", learnable by "scaling of preference modeling, where models do not cease learning despite variations in human preferences".
W5. Was not addressed in the response.
W6. The authors disagreed with the assessment of the reviewers and described some of their results. I find their arguments convincing.
W7. Eq.3 was further explained.


Overall, the paper has some aspects that might be of interest to the preference learning community; the scale of the dataset and experiments is certainly impressive. However, even considering the authors response, the positioning of the paper with respect to existing work remains poor. Also, concerns about the soundness methodology and annotator bias were not adequately addressed. The paper is thus not yet ready for publication.

**Reviewer Scores:**

I have no way of knowing know how the reviewers would have changed their scores.

---

### Decision · Program_Chairs · 2026-01-26

Reject